# Persistent representation of a prior schema in the orbitofrontal cortex facilitates learning of a conflicting schema

Ido Maor [1] ✉, James Atwell[1], Ilana Ascher[1], Yuan Zhao[2], Yuji K. Takahashi [1], Evan Hart[3], Francisco Pereira[2] & Geoffrey Schoenbaum [1] ✉

Schemas allow efficient behavior in new situations, but reliance on them can impair flexibility when new demands conflict. Evidence implicates the orbitofrontal cortex (OFC) in deploying schemas in new situations. But how does this role affect learning of a conflicting schema? Here we addressed this question by recording or transiently inactivating OFC neurons in rats learning odor problems with identical external information but orthogonal rules governing reward. OFC representations adapted to track the underlying rules, and both performance and encoding were faster on subsequent than initial problems. Surprisingly, when the rule changed, persistent representation of the prior schema predicted faster acquisition of the new, and disrupting OFC activity during initial schema learning, later impaired acquisition of the second schema. Thus, rather than interfering with new learning, OFC neural activity was linked to improved acquisition by preserving accurate representations of the prior schema alongside the new one.

Understanding the rules that govern a specific situation and generalizing them to other situations with similar structures is a fundamental cognitive ability essential for adaptive behavior[1–6]. This process acts as a mental shortcut, enabling efficient problem-solving through the application of preexisting templates constructed from prior experience in new situations[7–14]. The use of such templates, or schemas, is evident across a wide range of tasks and behaviors, where identifying key rules and extracting abstract principles leads to improved performance over time[15–19]. When you learn to drive, for example, you gradually assemble a rule set: how right-of-way works at junctions, what common signs mean, how speed limits are posted, when to yield, how to approach roundabouts, and where to keep your lane position. With practice, these pieces fuse into a schema that makes decisions fast and with low effort; your driving becomes smoother and more reliable, and the underlying schemas allow you to quickly adapt to new vehicles, road systems, or unfamiliar environments.

The OFC is thought to play a pivotal role within a circuit mediating the development and use of such schemas, specializing in the identification of hidden states—general principles or rules—that hold true across similar problems[20–22]. While the contribution of the OFC to this cognitive function is a relatively new proposal, its long-appreciated role supporting rapid reversal learning can be viewed as an example of this function[23–26]; as can the role of OFC in settings such as devaluation[27,28], in which a new goal value must be generalized to novel situations.

Yet, the application of prior knowledge can introduce incorrect assumptions or biases in new situations governed by conflicting rules. For example, when you go to a different country, driving conventions can shift: traffic may use the other side of the road, priority rules at junctions or roundabouts can change, similar-looking signs can carry different instructions, and speed limits may be in different units. Early on, your home-country schema intrudes, you check the wrong direction, treat a yield as a stop, misread a speed sign, or anticipate a maneuver that is not permitted. With exposure, the old mapping is suppressed, and a new rule set is adopted, restoring fluent performance under the new contingencies. While the driving example may

[1]National Institute on Drug Abuse Intramural Research Program, National Institutes of Health, Baltimore, MD, USA. [2]National Institute of Mental Health Intramural Research Program, National Institutes of Health, Bethesda, MD, USA. [3]Department of Psychology, University of Maryland, College Park, MD, USA. ✉e-mail: ido.maor@nih.gov; geoffrey.schoenbaum@nih.gov

be merely inconvenient and self-correcting with experience, the failure to adapt to the appropriate schema in new situations can result in much more serious ineffective and even maladaptive behaviors, which are most prominently exemplified in conditions like obsessive-compulsive disorder[29,30] and addiction[31,32].

If the OFC supports the formation and use of schemas, it becomes of interest how it manages orthogonal or conflicting behavioral schemas. One possibility is that the neural representation of a previously learned schema is silenced, overwritten, or jettisoned quickly when a new schema is encountered. Alternatively, the OFC might maintain parallel representations of both old and new schemas, allowing for flexible switching between them as environmental demands shift. This would allow the old schema to remain active to influence behavior, but might also hinder or prevent efficient adoption of the new schema.

Here, we adjudicated between these possibilities by inactivating or recording single-unit activity in the OFC of rats during learning of a series of odor problems in which external information was identical, but the rules governing reward were orthogonal. As expected, OFC representations adapted to track the underlying rules, and both performance and encoding were faster on subsequent than initial problems, consistent with schema formation. In rats trained on both rules, the OFC persistently maintained overlapping representations of the two schemas and, surprisingly, the strength of this interleaved representation correlated with the speed of acquisition of the new conflicting schema. Chemogenetic inactivation of OFC during learning of the initial schema impaired subsequent acquisition of the conflicting schema, suggesting a causal role for this correlative relationship.

## Results

### Behavioral performance during learning of orthogonal rules

Five male Long-Evans rats were trained on a series of odor-guided discrimination problems using two sets of eight unique odor cues (sets A and B) that predicted reward under one of two orthogonal rules (Fig. 1A). This initial cohort comprised males only due to technical constraints. We acknowledge this limitation; however, in a complementary inactivation experiment described later, both sexes were included, and no consistent main or interaction effects of sex were observed on task acquisition or performance (Supplementary Table 16). These findings are consistent with previous studies reporting no reliable sex differences in OFC-dependent cognitive functions[33,34].

Other than the difference in rules predicting reward, which we will describe below, sessions were otherwise identical in the structure of the events in each trial (Fig. 1B) and the distribution or potential rate of reward (0.51±0.01 for the first rule and 0.49±0.01 for the second rule). Each trial began with a light signaling the rat to sample an odor at the designated port. When a nosepoke into the port was detected, one of the eight odors was delivered, requiring the rat to decide whether to respond to a nearby fluid well to obtain a reward (Fig. 1B). Responses on positive trials resulted in the delivery of 50 microliters of sucrose solution, followed by a 4-second light-off period before the initiation of a new trial. Withholding a response, regardless of trial type, resulted in no outcome and terminated the trial.

During the initial training phase, the odor cues predicted reward based on a 'non-match' rule, where a reward was delivered if the response was to an odor different from the one sampled in the previous trial (Fig. 1C; "non-match rule"). In the first training session on this problem, the rats exhibited their default response of indiscriminately responding to the fluid well on all trials (Fig. 2A, 'Non-match A 1st'). But then, in the following sessions, they gradually learned to respond correctly on 'non-match' trials and withhold their response on 'match' trials. After 9–10 sessions, they learned to respond only if the odor was different from the odor sampled in the previous trial (Fig. 2A 'Non-match A last'). This response pattern resulted in increased behavioral accuracy based on the non-match rule (Fig. 2D,

left; blue line; ANOVA: F(9,40) = 10.95, p < 0.001, see Supplementary Table 1) and a decrease in the number of trials required to reach 80% accuracy for successive sessions (Fig. 2E, left; ANOVA: $F$(9,40) = 6.50, $p$ < 0.001). After showing robust and stable performance (80% correct for 3 consecutive sessions), the rats were trained on a new problem where the rule remained the same, but eight new odors were introduced (Fig. 2A, 'Non-match B'). They successfully generalized the match/non-match rule to the new odors, reaching 80% performance within a single session (Fig. 2D, E right). Finally, the rats were retested on the original problem, demonstrating robust and persistent retention of prior learning (Supplementary fig. 1- Non-match A').

After retesting on the original problem, rats began the second training phase, in which the odors remained the same, but the task rule changed to be based on 'cue-identity', where rewards were predicted by the identity of each odor rather than the match or non-match comparison with the prior trial (Fig. 1C; "cue-id rule"). Half of the odors (1–4) were associated with a potential reward ('rewarded odors') while the other half (5–8) were not ('non-rewarded odors'). The rats adapted their behavior to the new rule, learning to respond only to the rewarded odors and to withhold responses to the non-rewarded odors, regardless of the odor sampled in the previous trial (Fig. 2B, 'Cue-id A'). This adaptation was gradual, resembling initial learning of the match/non-match rule (Fig. 2G left; yellow solid line, Fig. 2H left), however, unlike that initial learning, errors were not committed randomly. Instead, there was a higher probability of responding when the presented odor was a non-match to the prior trial (Fig. 2I left; $F$(9,78) = 4.41, $p$ < 0.001). This resulted in high residual accuracy according to the non-match rule (Fig. 2G left; blue solid line), particularly in the beginning of each session (Supplementary Fig. 1). This pattern reemerged when the rats were presented with the next problem (Fig. 2B, 'Cue-id B'); they again initially followed the old rule before fully committing to the new relevant rule (Fig. 2G–I right). Notably, this pattern was different from that in the initial phase of learning on the non-match rule, where the rats did not show any bias in their errors based on cue-identity (Fig. 2D, yellow lines, $F$(9,40) = 0.97, $p$ = 0.48). Thus, their bias to follow the now irrelevant orthogonal 'non-match' rule, after a new 'cue-identity' rule was introduced, depended on the prior experience. To confirm this dependency, we trained another group of rats exclusively on the 'cue-identity' rule (Fig. 1A "control group"; $n$ = 4). These rats also learned to follow the 'cue-identity' rule (Fig. 2C), gradually increased their behavioral accuracy according to this rule (Fig. 2G, yellow dashed lines) and reached the learning criterion with fewer trials (Fig. 2H, dashed line). Their behavior was not different in any aspect from the behavior of the rats that underwent the full learning curriculum ($F$(9,67) = 0.64, $p$ = 0.75), except that their errors were distributed randomly, rather than being more likely when the odor was a non-match to the prior trial (Fig. 2I, horizontal lines, $F$(9,56) = 0.09, $p$ = 1.00). This pattern of errors resulted in a chance level accuracy according to the orthogonal non-match rule which was significantly lower than the accuracy of the main experimental group, particularly in the earlier stages of each problem (Fig. 2G, blue dashed lines, $F$(9,67) = 2.83, $p$ = 0.007). Overall, these results confirm that, when confronted with a new rule, the rats in our main experimental group showed residual effects of the prior training, which gradually diminished as they learned to apply the new schema appropriate to the new rule.

### Single unit correlates during learning of orthogonal rules

We recorded single units from the lateral orbitofrontal cortex (lOFC) during all of the training described above. The total number of neurons, number of neurons per rat, average firing rate, and percentage of responsive units remained stable throughout the training (Supplementary Fig. 2). However, the selectivity of individual neurons to different task components changed according to the rule in effect. While most of the neurons recorded in the first training session had a

**A.**

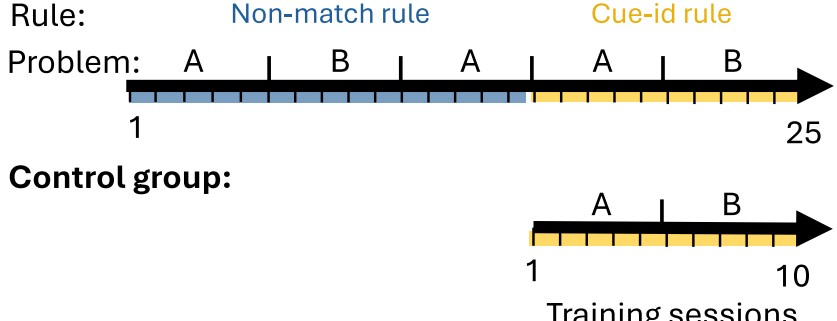

**B.**

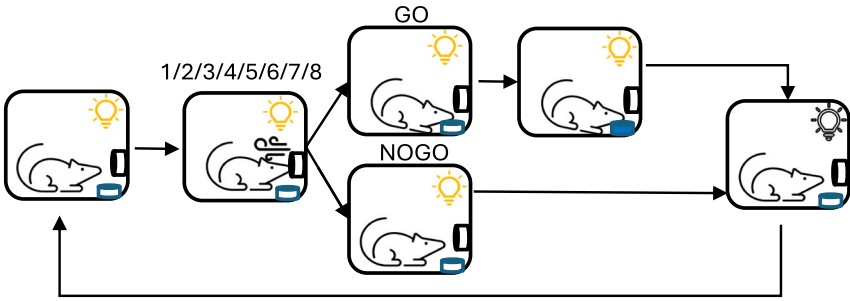

**C.**

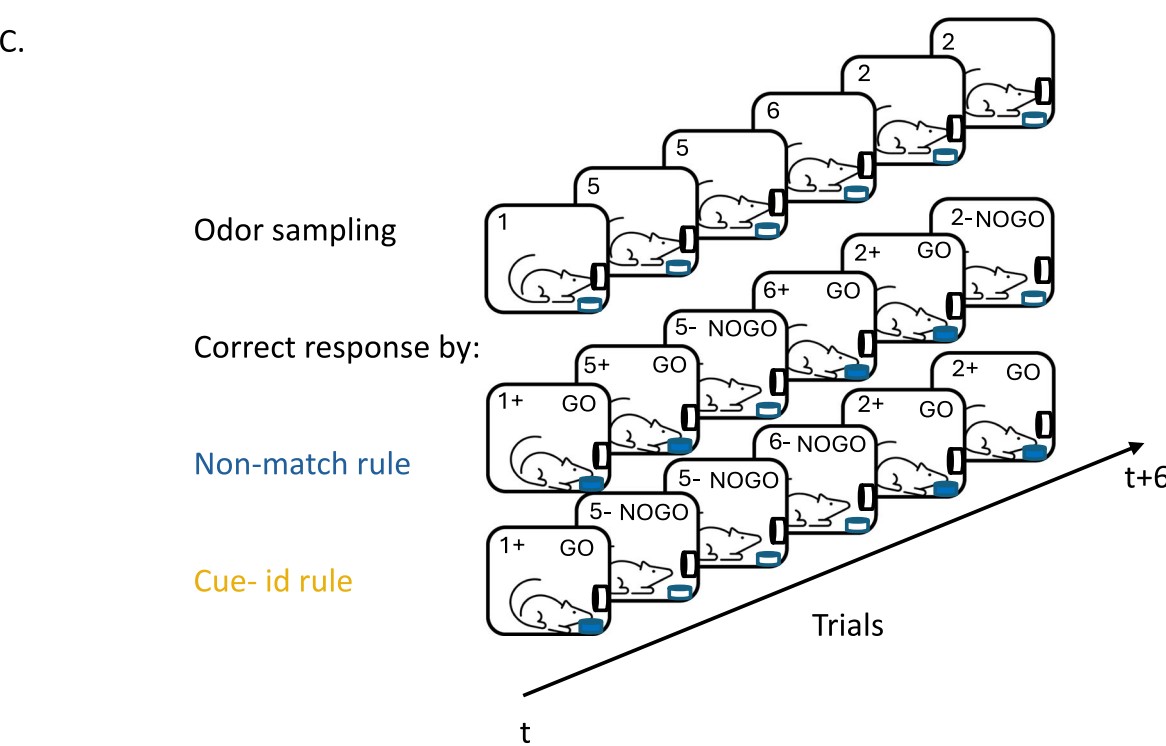

**Fig. 1 | Overview of task curriculum, trial structure, and rule design.**
**A** Curriculum overview. Rats were trained on a sequence of odor-guided discrimination tasks governed by two orthogonal rules: 'Non-match' and 'cue-id'. For each rule, two different odor sets, comprised of 8 different odors (problem A and B), were used for at least 5 consecutive sessions. Control group was trained solely on 'cue-id' rule. **B** Schematic of trial structure. **C** Schematic of the task design in 6 consecutive trials and the correct response according to the different rules. In each trial, one odor was presented (1–8), and the rat had to decide whether it was associated with a reward.

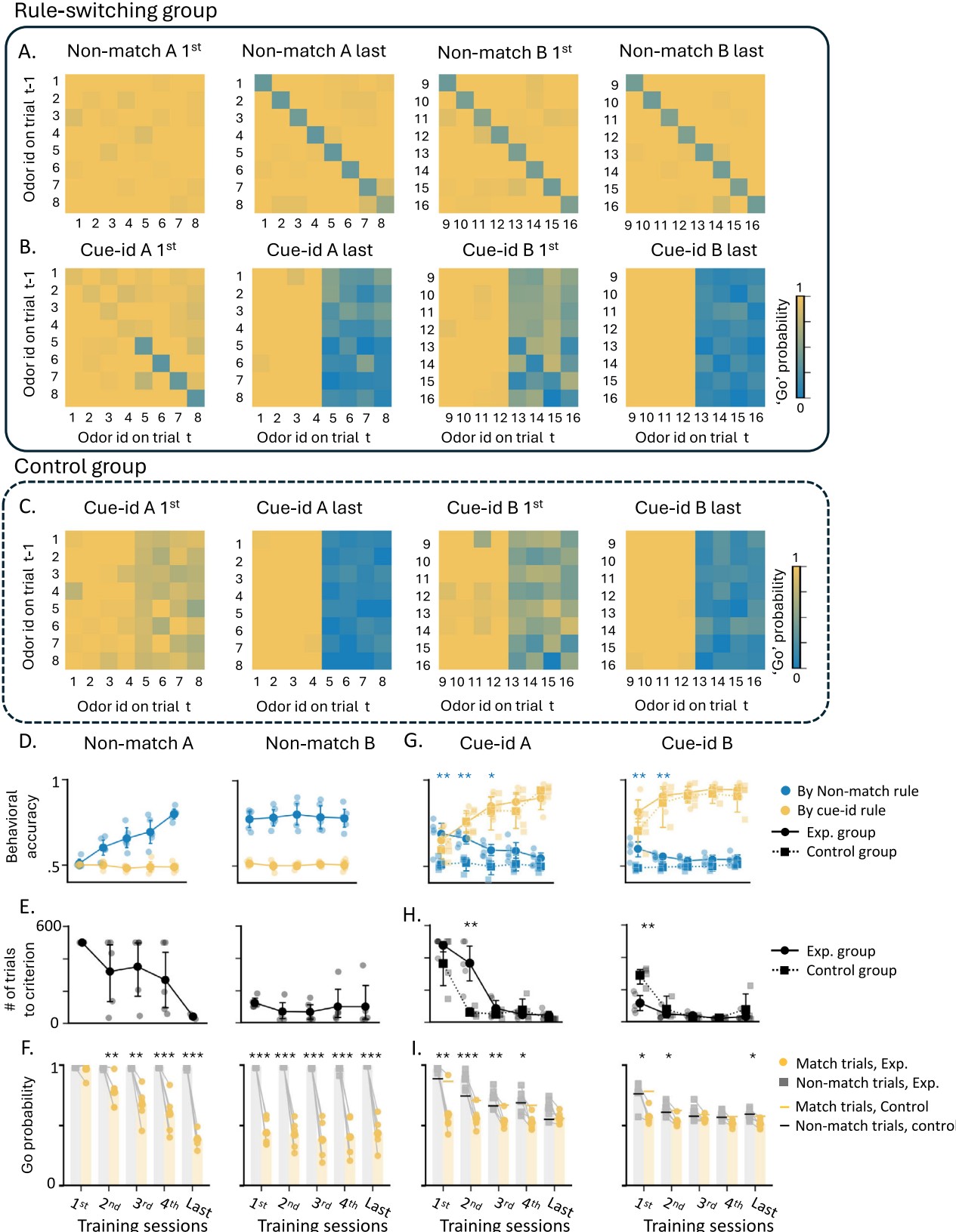

significant response during the odor sampling period (Supplementary Fig. 2D, Fig.3A, 'Non-match A, 1st'), only a few neurons fired differently in non-match versus match trials. This is captured in Fig. 3B, which shows the difference in the z-score versus baseline for non-match versus match trials for each unit (top) as well as the average z-score across all neurons (bottom). To quantify the proportion of neurons

with significant selectivity to the rule, we used the parameter-free ZETA-test[35] to compare the activity of each neuron in the odor sampling period in non-match and match trials (Fig. 3C). This analysis revealed that, in the first training session, less than 5% of the neurons were selective to the non-match rule. However, as learning progressed and the rule consolidated, more neurons had a different response in

**Fig. 2 | Learning dynamics under non-match and cue-identity rules.**
**A** Probability of responding 'go' across all trial types (different odors in match/non-match configurations) in the first and last training sessions under the non-match rule for (**A**) and (**B**) problems. **B** Same as (**A**) but for training under the cue-identity rule. **C** Same as (**A**, **B**), but for the control group, which was trained exclusively on the cue-identity rule. **D** Accuracy under the non-match rule (blue) and cue-identity rule (yellow) in problem (**A**) (left) and (**B**) (right). Data in this and all other panels are shown as mean ± SEM with individual data points overlaid (n = 5 rats unless otherwise indicated). Accuracy increased significantly across sessions for the non-match rule (one-way ANOVA: $F_{(9,40)} = 10.95$, $p = 2.3 \times 10^{-8}$) but not for the cue-identity rule ($F_{(9,40)} = 0.97$, $p = 0.48$; Supplementary Table 1). **E** Number of trials to reach learning criterion under the non-match rule in (**A**) and (**B**) problems. Trials decreased significantly across sessions (one-way ANOVA: $F_{(9,40)} = 6.50$, $p = 1.2 \times 10^{-5}$). **F** Go probability under the non-match rule separated into match (yellow) and non-match (gray) trials. A two-way ANOVA revealed main effects of session and

condition, with a session × condition interaction ($F_{(9,80)} = 10.90$, $p = 7.1 \times 10^{-11}$). Post hoc Tukey tests confirmed differences between trial types for all sessions except the first (*$p < 0.05$, **$p < 0.01$, ***$p < 0.001$). **G** Accuracy under the non-match and cue-identity rules in (**A**) and (**B**) problems in experimental group (solid; n = 5 rats) and controls (dashed; n = 4 rats). For the non-match rule, a session×group interaction emerged (two-way ANOVA: $F_{(9,67)} = 2.83$, $p = 0.007$), driven by early-session differences ($p < 0.02$). No group effects were observed for the cue-identity rule. **H** Number of trials to criterion under the cue-identity rule in Problems (**A**) and (**B**) for experimental and control groups. A session×group interaction was observed (two-way ANOVA: $F_{(9,67)} = 7.78$, $p = 9.7 \times 10^{-8}$), with significant differences in session 2 of Problem A and session 1 of Problem B. **I** Go probability under the cue-identity rule in Problems (**A**) and (**B**) for experimental (bars; n = 5) and control (horizontal lines; n = 4) groups. In the experimental group, session, condition, and their interaction were significant (two-way ANOVA: $F_{(9,78)} = 4.41$, $p = 0.0001$). In controls, only a session effect was significant ($F_{(9,56)} = 9.77$, $p = 8.3 \times 10^{-9}$).

non-match and match trials (Fig. 3A, B, 'Non-match A, last', paired $t$ test: $t_{(205)} = 2.56$, $p = 0.011$; Supplementary Table 3), and the proportion of the non-match rule selective neurons gradually increased to 25% (Fig. 3C, 'Non-match A'). This increased representation persisted as the schema generalized to a new problem with a different odor set (Fig. 3A–C; 'Non-match B', $t_{(205)} = 3.58$, $p < 0.001$).

To evaluate the degree of selectivity to the orthogonal 'cue-identity' rule during match/non-match sessions, we compared the difference in z-scores of individual units, as well as the proportion of selective neurons based on the ZETA-test comparison, dividing the trials into rewarded and non-rewarded trials according to both rules (i.e. either the current match/non-match or the future cue-identity rule). The proportion of neurons selective to the future 'cue-id rule' was relatively low, declined in the first few sessions, and then remained low for the remainder of training on the 'non-match rule' (Fig. 4A, B, 'Non-match A').

When cue-id training began, OFC activity gradually shifted to represent the new relevant rule (Fig. 4C, D, Supplementary Fig. 3). The difference in response to rewarded versus non-rewarded trials according to this rule was significant in many of the units (Fig. 4C, blue markers), and the proportion of neurons selective to this rule gradually increased (Fig. 4D, dark yellow bars). However, a significant proportion of neurons still exhibited a substantial difference in response to rewarded versus non-rewarded trials according to non-match rule (Fig. 4C) and remained selective to this irrelevant rule (Fig. 4D, dark blue and gray bars; 5%). This residual selectivity was observed even after many sessions and after behavior fully conformed to the new rule (Fig. 2G).

To confirm that this residual representation of the irrelevant rule was not due to intrinsic "mixed selectivity" or encoding of latent relationships, instead reflecting prior experience with the first rule, we compared it with the neural representation in the control group of rats that trained solely on the cue-id rule. In these rats, the difference in responses to rewarded versus non-rewarded trials according to the non-match rule was significantly smaller than those of the 'rule-switching' group (Fig. 4C, gray markers; Kolmogorov–Smirnov test, chosen because it compares entire distributions rather than just means: $D = 0.214$, $p = 0.026$; Supplementary Table 4) with almost no units that were significantly selective to this irrelevant rule (Fig.4d, light blue and white bars, $F_{(1,67)} = 14.46$, $p < 0.001$ and $F_{(1,67)} = 6.99$, $p = 0.01$, for the proportion of units responsive to the non-match rule or to both rules, respectively; Supplementary Table 5), thus suggesting that prior behavioral schema implementation had a prolonged effect on the OFC representation.

## Patterns of population activity during learning of orthogonal rules

To investigate the neural representation of conflicting behavioral schemas by the population activity in the OFC, we reduced the dimensionality of the neural responses using Uniform Manifold

Approximation and Projection (UMAP), embedding the data into a three-dimensional space. The UMAP embeddings were plotted to provide a clear visualization of trial-specific neural representations, with data points distinguished by odor identity (color) and trial configuration (marker style). As rats learned to implement the first rule, the neural representation evolved to form distinct clusters of match and non-match trials reflecting the rule in effect (Fig. 5A right; 'x' and 'o' markers, respectively), rather than on the identity of the odor presented (different colors). To further quantify this separation, dendrograms constructed from the UMAP embeddings demonstrated strong clustering of trials by reward contingency under the 'non-match' rule (Fig. 5A right). When the same rule was generalized to the next problem, a similar population representation was observed, with trials continuing to show clear separation according to the non-match rule (Supplementary Fig. 4). This consistency across problems highlights the stability of the neural representation when the task rule remains unchanged and the unimportance of cue identity in rats first trained on the non-match rule.

To quantify how these representations evolved with training, we measured the distances between clusters in the UMAP space while applying a leave-one-subject-out (LOSO) analysis to verify robustness across animals. For each session, embeddings and centroid distances were recomputed while systematically excluding one rat at a time, and the resulting values were averaged across iterations. This analysis revealed that the separation between odors in the match versus non-match configurations increased across sessions, reflecting consolidation of the first behavioral schema (Fig. 5D, blue curves). Significant separation relative to shuffled data (blue asterisks) was observed in all sessions except the first (adjusted $p < 0.05$, permutation test, 1000 iterations, see Supplementary Table 6).

When rats were confronted with the new cue-identity rule, neural population activity in OFC adapted to separate trials according to this rule. Across these sessions, UMAP embeddings showed a growing separation between rewarded and non-rewarded odors (Fig. 5B, yellow shades; Fig. 5E, solid yellow lines). These distances were significantly greater than those from permutation-based null distributions in every session ($p < 0.05$). However, the neural representation also retained the previously learned 'non-match' rule, as evidenced by the partial separation of trials based on the irrelevant rule (Fig. 5B, "o" vs. "x" markers; Fig. 5E, solid blue lines). Distances for the non-match rule were also significantly above permutation null values across all sessions ($p < 0.05$).

To confirm that the persistence of the old rule's representation was not due to chance and indeed reflected prior experience with the non-match rule, we compared these findings to those in the control group trained exclusively on the cue-identity rule. In these rats, the UMAP analysis revealed a robust separation between rewarded and non-rewarded odors, similar to the separation observed in the experimental group (Fig. 5C; Fig. 5E, dashed yellow lines, significant

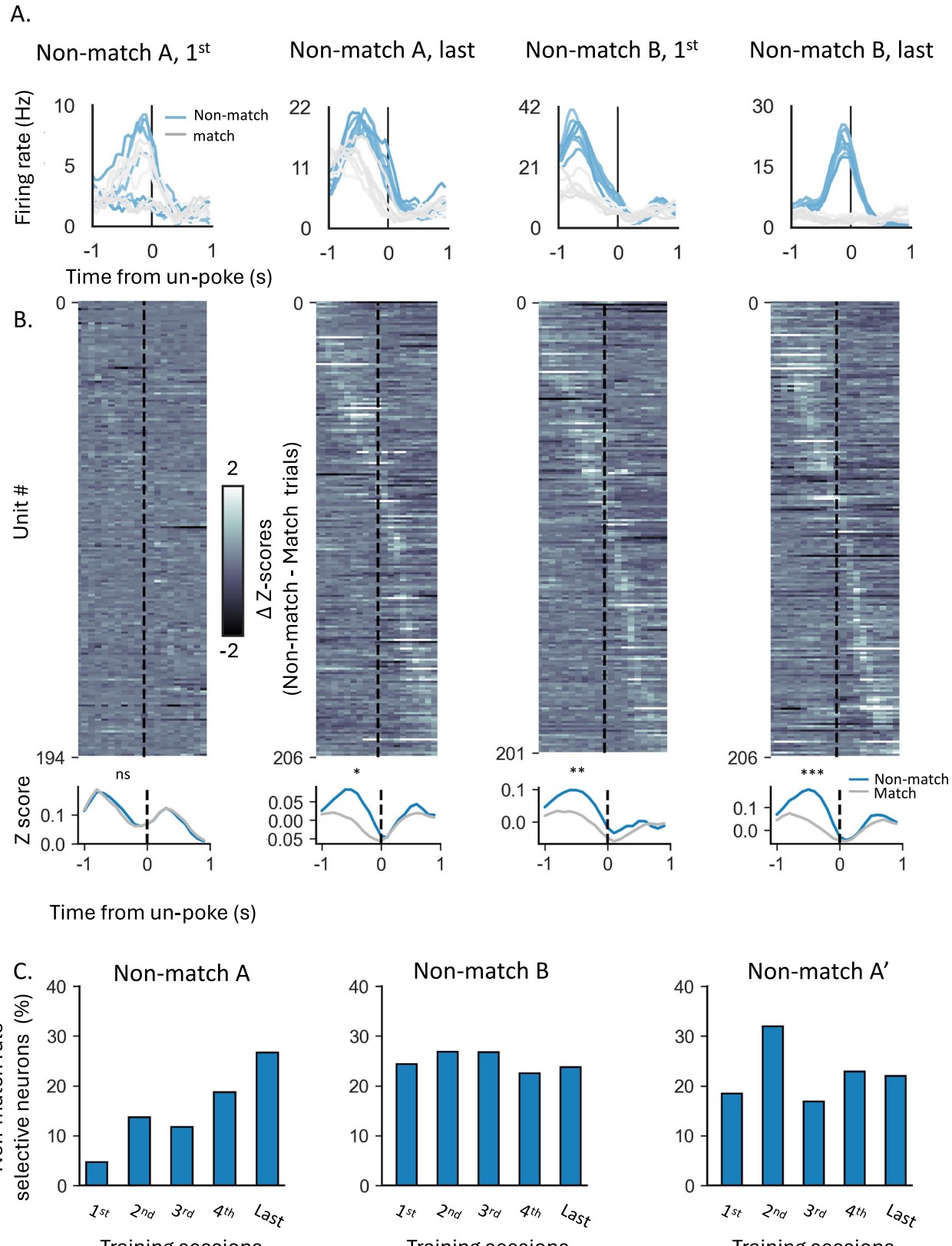

**Fig. 3 | Increased single unit selectivity to the learned non-match rule.**
**A** Peristimulus time histograms (PSTHs) of example neurons in response to the different odors in rewarded (blue) and non-rewarded (gray) trials. The PSTHs were aligned to the decision time (un-poke from odor port). **B Top**. Difference in Z-score of all recorded units per session in rewarded versus non-rewarded trials, aligned to the decision time. Units were subsequently ordered by their latency to peak activity, and the colormap was trimmed between –2 and 2 to enhance visibility. **Bottom**. Z-score in rewarded (blue) and non-rewarded (gray) trials, averaged over all units. Asterisks indicate significance levels of paired t-tests, comparing the mean firing rate of individual neurons during the odor sampling period between non-match and match trials (ns- non-significant, *$p < 0.05$, **$p < 0.01$, ***$p < 0.001$; $p = 0.3$; $p = 0.011$; $p = 0.004$; $p = 0.0004$; See Supplementary Table 3). **C** Proportions of neurons with a significant selectivity to the non-match rule, based on parameter-free ZETA-test (see methods) for problem non-match A, problem non-match B, and the retest phase of non-match A (A').

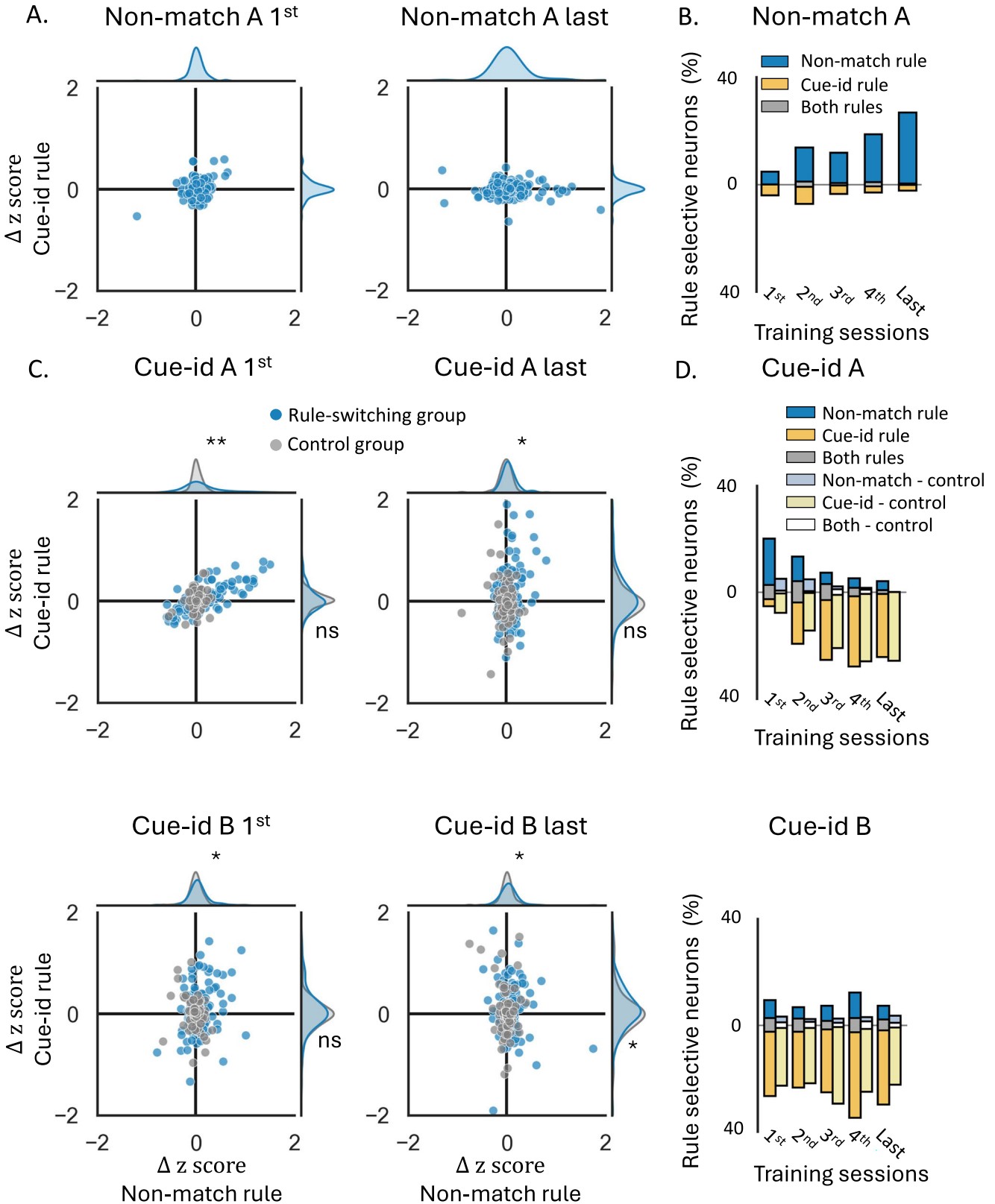

differences from shuffled data in all sessions but the first, $p < 0.05$). However, the representations of match and non-match trials remained overlapping and were not different from shuffled datasets (Fig. 5E, dashed blue lines).

The exclusivity of the residual representation of the prior rule to the experimental group was further confirmed by directly comparing the degree of separation between experimental and control animals against the distribution of differences obtained from two shuffled datasets. This analysis showed that, in most sessions, the experimental group exhibited significantly greater separation according to the non-match rule compared to controls (see Supplementary Table 7). Together, these results indicate that OFC simultaneously carries the relevant cue-identity code and a residual non-match code.

**Fig. 4 | Increased single unit representation of the new rule with residual encoding of the irrelevant rule. A** Delta z-score of individual units in rewarded vs. non-rewarded trials, plotted for the non-match rule (X-axis) and the cue-identity rule (Y-axis), during the first (left) and last (right) sessions of problem A under the non-match rule. Marginal axes display the distributions of delta z-scores for each rule. **B** Proportions of neurons showing significant selectivity, for the non-match rule (blue), cue-identity rule (yellow), or both rules (gray) across different sessions of problem A under the non-match rule. **C** Same as (**A**), but for problems (**A**) (top) and (**B**) (bottom) under the cue-identity rule. Blue markers represent individual units from rats that underwent the full learning curriculum and first learned the non-match rule, while gray markers correspond to units from the control group, which was trained only on the cue-identity rule. Marginal axes show the distributions of delta z-scores for the rule-switching group (blue) and control group (gray). (*$p < 0.05$, **$p < 0.01$, Kolmogorov–Smirnov test comparing the distributions of delta z-scores in the two groups; see Supplementary Table 4). **D** Same as (**B**), but for sessions under the cue-identity rule, comparing the rule-switching group (dark bars) and control group (light bars). A two-way ANOVA revealed a significant main effect of group for the non-match rule responsive units ($F_{(1,67)} = 14.46$, $p < 0.001$) and for the both rule responsive units ($F_{(1,67)} = 6.99$, $p = 0.010$), indicating that the control group had significantly fewer units that were selectively responsive to this irrelevant rule compared to the rule-switching group. In contrast, there was no significant difference in the proportion of units that were selective for the relevant cue-identity rule ($F_{(1,67)} = 0.04$, $p = 0.841$; see Supplementary Table 5).

We next asked how these two codes are arranged in the neural population space. At the single-neuron level, only a small fraction of units multiplexed both rules (Fig. 4D, gray bars); most neurons signaled rewarded versus non-rewarded trials for one rule or the other. To assess whether the population nevertheless encodes the two rules in separable versus aligned subspaces, and to characterize how this geometry evolves with learning, we quantified the cross-condition generalization performance (CCGP) for each rule and the angle between the corresponding population axes[36]. Consistent with the idea that the geometry of the neural representation supports abstraction, CCGP asks whether a decoder trained on a subset of odor conditions within a contingency generalizes to held-out odors from the same contingency.

During acquisition of the non-match schema, CCGP based on this rule increased from 0.59 to 0.96 with a strong session effect (Fig. 5f, blue, one-way ANOVA: $F_{(9,40)} = 122.43$, $p = 3.06 \times 10^{-26}$), indicating that the population encoded the abstract non-match rule rather than memorizing odor sequence; in the same sessions, cue-id-CCGP was around chance, as expected before learning this rule (Fig. 5F, yellow). After the rule switch, OFC population geometry reorganized to support decoding based on the new relevant rule: cue-id-CCGP rose rapidly and remained high in the experimental group (Fig. 5G, yellow solid; 0.65–0.97; one-way ANOVA: $F_{(9,39)} = 127.13$, $p = 4.6 \times 10^{-26}$), indicating a rapid shift to an odor-invariant code that generalized to held-out odors with the same potential outcome. Importantly, non-match-CCGP stayed above chance during all sessions (Fig. 5G, blue solid; 0.61–0.82; $F_{(9,39)} = 5.05$, $p = 1.6 \times 10^{-4}$), demonstrating again the persistent encoding of the prior-rule alongside the new rule. Controls trained exclusively on the cue-id rule showed similarly robust cue-id-CCGP (Fig. 5G, dashed yellow; all sessions were significantly higher than chance; $F_{(9,28)} = 26.34$, $p = 2.2 \times 10^{-11}$), but non-match-CCGP remained near chance (Fig. 5G, dashed blue; 0.53–0.57; $F_{(9,28)} = 1.92$, $p = 0.091$), confirming that residual abstract rule coding depends on prior experience. Geometrically, the two rules readout axes drifted, with learning of the new rule, toward near-orthogonality (Fig. 5I, solid lines; cosine session effect $F_{(9,39)} = 29.85$, $p = 8.46 \times 10^{-15}$), indicating increasingly separable subspaces for the two conflicting schemas. Overall, the OFC population geometry supported parallel, rule-specific abstractions: the task-relevant cue-identity code was expressed with high generalization, while a residual non-match code remained concurrently readable in a partly independent subspace.

The similarity in the patterns of activity in the neural activity space on different trial types can also be represented in matrix form, which can be used to understand how reliably information about different aspects of the task is represented, in this case, the two rules. To illustrate this, we constructed three exemplar templates showing similarity based on each rule alone or in combination (Fig. 6A). In the 'non-match template', similarity is high for odors presented in non-match configuration or for odors presented in the match configuration, regardless of cue identity, whereas in the 'cue-id template', similarity is high for rewarded odors or non-rewarded odors based on identity, regardless of whether they were presented in match or non-match configuration. Finally, in the 'both-rules template', the similarity of the activity pattern is high between rewarded odors or between non-rewarded odors, but only if they also share the same match/non-match trial configuration. We compared those templates to the results from an analysis of neural population firing rates during the odor sampling time during learning across sessions involving the two rules. To quantify the response similarity between different trial types, we employed a Support Vector Machine (SVM) decoder, trained to predict the trial type based on a vector of firing rates for the neurons in the population, for each rat and session separately. We utilized a leave-one-out cross-validation strategy to assess the accuracy of the decoders and plotted its predictions as confusion matrices (Fig. 6B–D; Supplementary Fig. 5). As rats learned to implement the first rule, the decoders increasingly confused trial types that shared the same potential outcome (match/non-match; Fig. 6B) and became similar to the 'non-match template' (Fig. 6E, blue lines, see methods). In contrast, the similarity to the 'cue-id template' or to the 'both-rules template', remained low (Fig. 6E, yellow and gray lines).

When confronted with the new rule, the pattern of neural activity changed to reflect the outcome according to the new rule (Fig. 6C) and became better aligned with the template corresponding to the 'cue-id' rule (Fig. 6F, solid yellow lines). However, the similarity between trials that predicted reward by both rules was higher (Fig. 6C). Consequently, while the similarity to the 'cue-id template' increased, the similarity to the 'non-match template', which was based on the former rule, or to the 'both-rules template' reflecting the integration of the two rules, remained significantly above chance over many sessions involving both problem A and problem B (Fig. 6F, solid blue and gray lines).

This result contrasts with the findings in the control group of rats, where the population activity in OFC converged, with training, to become better aligned with the 'cue-id' template (Fig. 6D, Fig. 6F; dashed yellow lines, no difference in similarity to this template was observed between the control group and the main experimental group, as indicated by a two-way ANOVA: $F_{(1,35)} = 0.06$, $p = 0.811$; see Supplementary Table 9). By contrast, similarity to the irrelevant match–non-match rule and both-rules templates was significantly lower in the control group compared to the experimental group (Fig. 6F, dashed blue and gray lines). These statistical results apply to both plots shown in Fig. 6F and indicate significant group effects for the non-match ($F_{(1,35)} = 14.28$, $p < 0.001$) and both-rules ($F_{(1,35)} = 5.61$, $p = 0.024$) templates. This difference is most striking in a direct comparison of classification along the main and the side diagonals of the confusion matrices after cue-identity training in both groups, which were similar in controls but asymmetric in the experimental group (Fig. 6G, Mann–Whitney U test, $U = 1322$, $p < 0.001$). This difference was not due to differences in performance observed during the very first sessions of cue-identity training, as excluding these sessions from both groups still yielded a significant group difference ($U = 766$, $p = 0.029$). Analysis of neural decoders that trained to predict reward according to each rule further confirmed that the residual

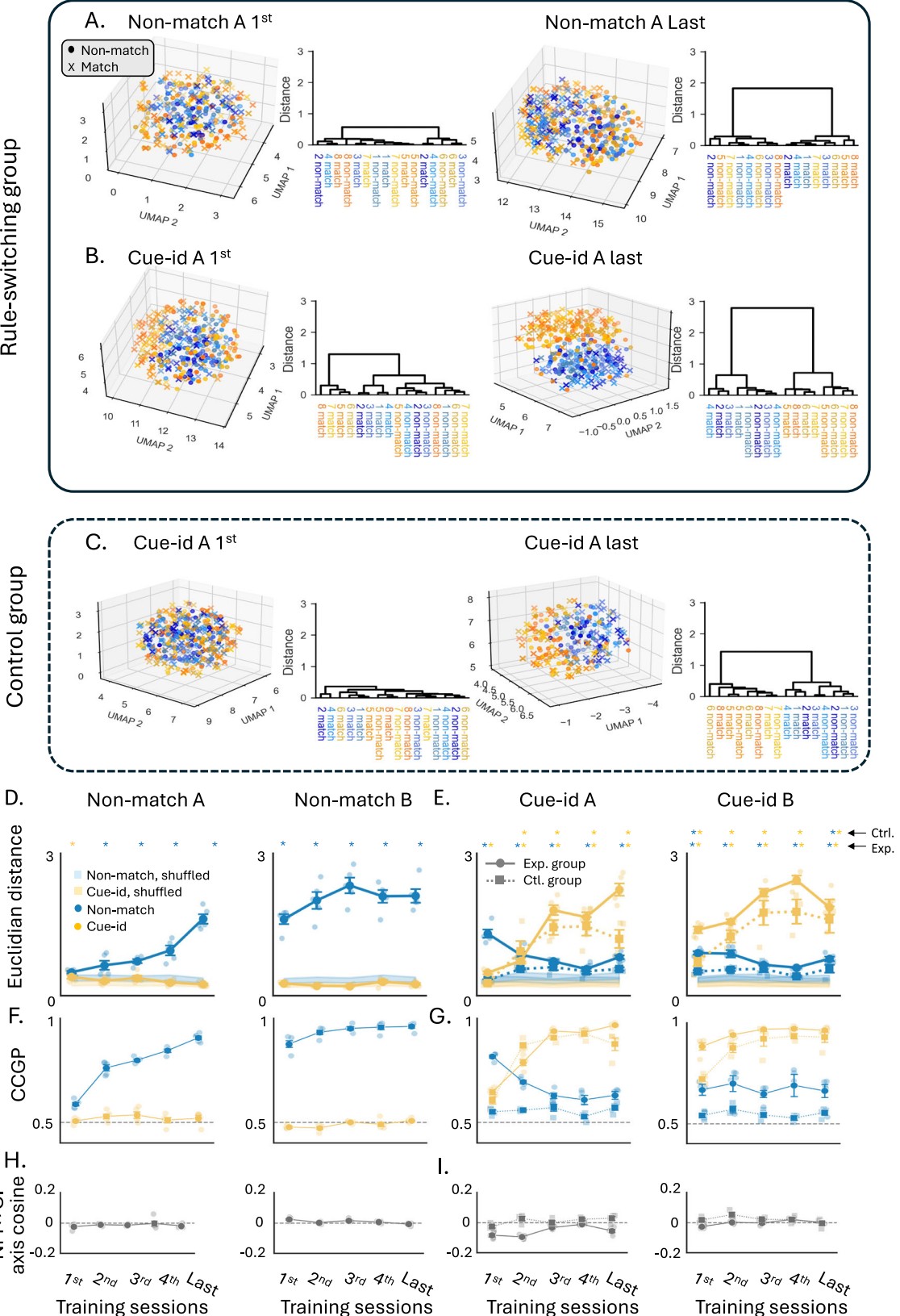

**Nature Communications** | (2026)17:2610

representation of the irrelevant rule was unique to the group of rats who previously learned that rule (Supplementary Fig. 6a) and was not an artifact of the animal selection (Supplementary Fig. 6b).

Overall, these findings, derived using multiple approaches, converged on the same conclusion. First, both individual neurons in the OFC and the pattern of activity across ensembles of OFC neurons dynamically adapt to encode the relevant behavioral schema. Second, they do so while retaining significant traces of the previously acquired rule, even long after behavior had fully conformed to the new rule across multiple problems.

**Fig. 5 | Increased neural population representation of the new rule with persistent and orthogonal representation of the irrelevant rule. A** 3D UMAP visualization of neural population activity during the odor sampling, with trials separated by odor identity (blue: odors 1–4, orange: odors 5–8) and trial configuration ('o': non-match, 'x': match). Each raster represents pseudo-population responses from the first and last sessions of non-match A problem. Adjacent dendrograms show trial-type clustering based on UMAP distances. **B** Same as (**A**), for the cue-identity A problem. **C** Same as (**B**), for the controls. **D** Euclidean distances in UMAP space between odors in match vs. non-match configurations (blue) and rewarded vs. non-rewarded groups (yellow) across sessions under the non-match rule. Values were computed using a leave-one-subject-out (LOSO) analysis, recomputing embeddings while excluding one rat per iteration. Data in this and all subsequent panels are shown as mean ± SEM with LOSO points overlaid; $n = 5$ LOSO iterations for experimental group and $n = 4$ for controls. Shaded bands show corresponding distances from shuffled datasets. Significance was assessed by comparing LOSO means to permutation nulls (1000 shuffles; Bonferroni corrected). Significant effects for the non-match rule (blue asterisks) were observed in all

sessions except the first; for the cue-identity rule (yellow asterisks), only the first session was significant (see Supplementary Table 6). **E** Same as (**D**), but comparing the rule-switching group (solid) and controls (dashed) under the cue-identity rule; see Supplementary Table 6. **F** cross-condition generalization performance CCGP during non-match sessions. Non-match-CCGP increased from 0.59 to 0.96 (one-way ANOVA: $F(4,40) = 122.43$, $p = 3.1 \times 10^{-26}$; see Supplementary Table 8), with all sessions above chance (permutation tests). Cue-id-CCGP remained near chance. **G** Same as (**F**) but comparing experimental and control groups during cue-identity sessions. Experimental rats showed rapid increases in cue-identity CCGP ($F(9,39) = 127.13$, $p = 4.6 \times 10^{-26}$; all sessions > 0.5). Non-match-CCGP persisted above chance (0.61–0.82; $F(9,39) = 5.05$, $p = 1.6 \times 10^{-4}$; all sessions > 0.5). Controls showed robust Cue-id-CCGP (all sessions > 0.5; $F(9,28) = 26.34$, $p = 2.2 \times 10^{-11}$) but Non-match-CCGP near chance. **H** non-match rule ↔ Cue-Identity rule (NM ↔ CI) axis cosine during non-match sessions. **I** Same as (**H**) for cue-identity sessions, comparing experimental and control groups. Cosine values were slightly negative early and moved toward 0 with learning ($F(9,39) = 29.85$, $p = 8.46 \times 10^{-15}$), indicating increasing subspace separation.

## Effects of residual encoding on optimal behavior during learning of orthogonal rules

Interleaving old information with new can allow the old information to remain available to be used in future scenarios, however this practice is generally assumed to have contrary effects on behavior. For example, the representation of previously acquired rules would be expected to interfere with successful acquisition and expression of a new conflicting or even orthogonal rule. If this is true, we would expect to find a positive correlation between encoding of the previous rule and the tendency of the rat to follow that irrelevant rule. Alternatively, it is also possible that the network is able to represent the previous rule in parallel with the new learned rule, without any direct effect on the behavior or the acquisition of the new rule. Or, finally, it is possible that representation of the previous rule is beneficial to acquisition of the new. Indeed, there are even some examples, such as the over-training reversal effect[37,38], which suggest that strong representation of a prior contradictory rule can facilitate new learning.

To test between these possibilities, for each rat and each session, we computed the correlation between (1) the accuracy of the neural decoder at distinguishing between potential outcomes based on the two rules, and (2) the accuracy of behavior according to each rule. In the first learning phase, when only one rule was available, behavioral accuracy was significantly correlated with decoder accuracy based on the relevant non-match rule (Fig. 7A, blue markers; Problem A: $R^2 = 0.60$, $p < 0.001$, $\beta = 0.76$, $p < 0.001$; Problem B: $R^2 = 0.52$, $p = 0.0003$, $\beta = 1.03$, $p < 0.001$; see Supplementary Table 10). The better the decoder distinguished between match and non-match trials, the more accurately the rats followed that rule. During this phase, the correlation with the cue-identity rule was weak or non-significant (Fig. 7A, yellow markers; Problem A: $R^2 = 0.44$, $p = 0.0014$, $\beta = -0.33$, $p = 0.001$; Problem B: $R^2 \approx 0.00$, $p = 0.985$, $\beta = 0.002$, $p = 0.985$).

When confronted with the new cue-identity rule in the second phase of training, the first two sessions were characterized by a positive correlation between the accuracy of the neural decoder and of the behavioral response, for each of the two rules. As learning progressed and the two rules were distinguished by behavior (Fig. 2), the correlation between neural decoding and behavioral performance of the new relevant rule remained positive (Fig. 7B, yellow markers; Problem A: $R^2 = 0.88$, $p < 1 \times 10^{-9}$, $\beta = 0.96$, $p < 0.001$; Problem B: $R^2 = 0.37$, $p = 0.006$, $\beta = 0.45$, $p = 0.006$; see Supplementary Table 10). By contrast, correlations with the irrelevant non-match rule inverted: the better the decoder distinguished between rewarded and non-rewarded trials according to this rule, the less the rats followed it in their responses (Fig. 7B, blue markers; $R^2 = 0.805$, $p < 0.001$, $\beta = -0.98$, $p = 0.001$; see Supplementary Table 10). This shift was also reflected in the per-session slope analyses (Supplementary Table 11): early sessions

showed steep positive slopes between decoder and behavior (sessions 1–2: slopes = 0.89, 0.37; $r = 0.86$, 0.95), but these flattened to near zero (session 3: slope = 0.03, $r = 0.06$) and inverted in later sessions (sessions 4–5: slopes = -0.46, -0.34; $r = -0.58$, -0.87). On average, slopes were positive early (mean = 0.63) but negative late (mean = -0.26), and this difference was supported by a significant Fisher's Z test ($z = 4.20$, $p = 2.7 \times 10^{-5}$; Supplementary Table 11).

This inverse relationship in the later stages was not observed in the control group of rats who were not trained on the non-match rule (Fig. 7C, blue markers; Problem A: $R^2 = 0.02$, $p = 0.599$, $\beta = -0.12$, $p = 0.599$; Problem B: $R^2 = 0.001$, $p = 0.916$, $\beta = -0.018$, $p = 0.916$; see Supplementary Table 10).

To directly investigate the relationship between the neural representation of the irrelevant rule and the behavioral response, on a trial-by-trial basis, we focused on the incongruent trials where reward is expected according to the non-match rule but not according to the new rule. For these trials, we calculated the proportion of trials that were correctly labeled as rewarded trials by the non-match neural decoder, but where the rats nevertheless chose not to respond, in accordance with the new relevant cue-identity rule. This probability of ignoring the 'go' prediction of the old rule increased as training on the new rule progressed, and, surprisingly, was positively correlated with the fidelity of the neural representation of the old rule (Fig. 7D, $R^2 = 0.27$, $\beta = 1.14$, $p = 0.002$, $F(1,32) = 11.98$) such that the fidelity of the neural representation of the old rule became a good predictor of the behavioral accuracy according to the relevant cue-id rule (Fig. 7E, $R^2 = 0.34$, $\beta = 0.40$, $p < 0.001$, $F(1,32) = 16.67$). In other words, the more robust and faithful the representation of the old irrelevant rule, the more expert the rat became at ignoring its prediction and acting in agreement with the prediction of the new relevant rule. Notably, this tendency to better acquire the new rule cannot be explained by a superior learning ability, as no significant correlation was found between performance on the new cue-id rule and performance on the initial non-match rule (Fig. 7F; Pearson correlation, all $p > 0.1$). Thus, animals that learned the first rule more efficiently did not necessarily acquire the second rule faster. While these results do not establish causality, they suggest that persistent representations of prior schemas in OFC may actually facilitate acquisition of new, contradictory schemas.

If persistent representation of a prior schema in OFC contributes to more efficient learning of a new, contradictory schema, as suggested by the observed correlations, then having OFC "online" during acquisition of the initial schema should be necessary for normal acquisition of the second, contradictory schema. Thus, in our setting, inactivating OFC during non-match learning should slow the subsequent transition to the cue-identity rule. Conversely, if

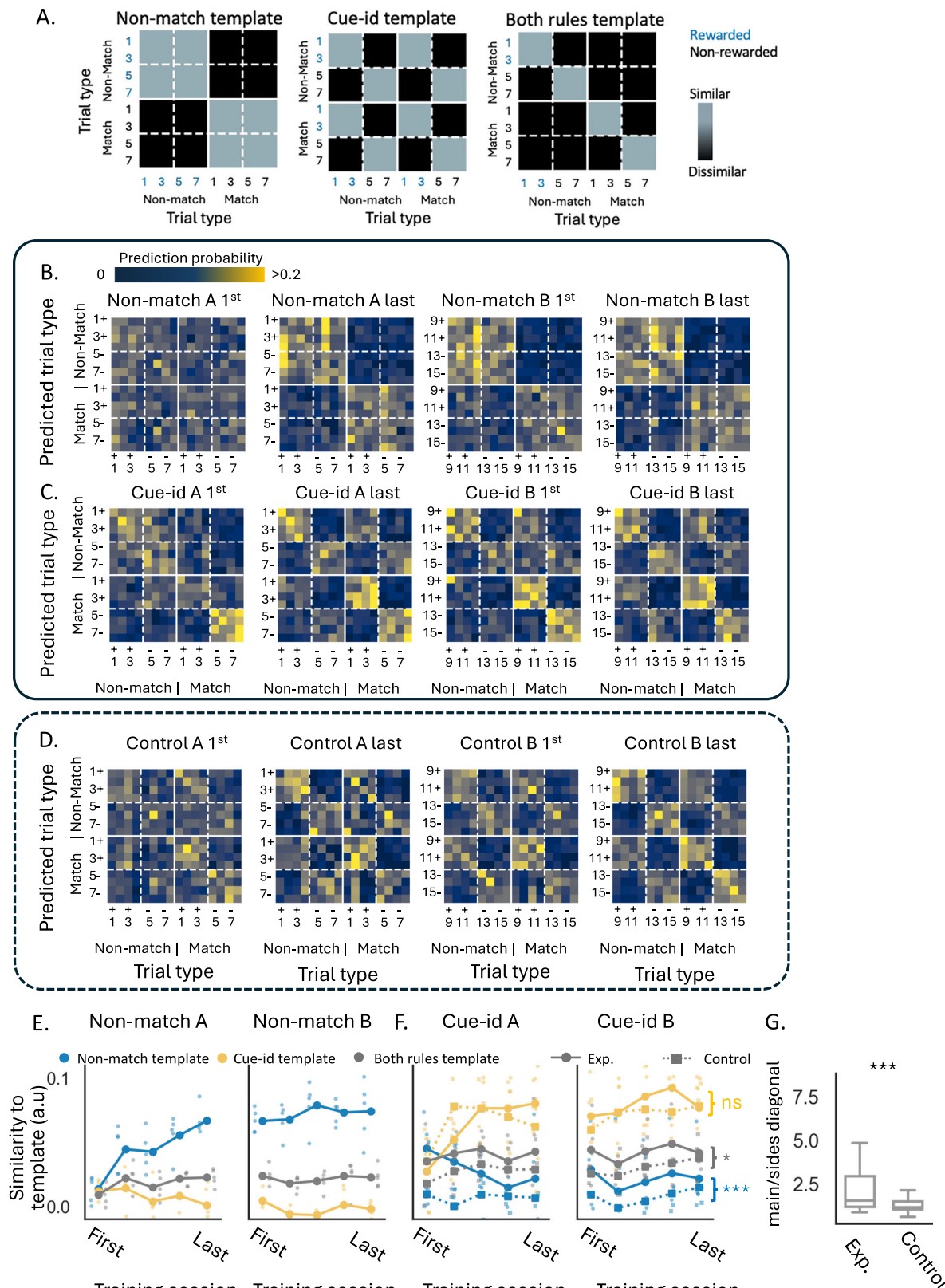

representation of the prior rule in OFC interferes with new learning (the more conventional assumption), then inactivating OFC during non-match learning should accelerate the transition to the cue-identity rule. To test between these competing predictions and directly assess the mechanistic role of OFC, we chemogenetically inactivated the OFC in a new group of rats during learning of the non-match rule and prior

to the switch to the new, conflicting cue-identity rule. For this, male and female rats were transfected with either hM4D(Gi)-mCherry (inhibitory DREADD receptor; *n* = 5) or mCherry alone (control; *n* = 5) in the lOFC (Fig. 8A). No consistent main or interaction effects of sex were observed (*F*(1,240) = 3.78, *p* = 0.053; Supplementary Table 16). After a two-week period to allow for viral expression, animals

**Fig. 6 | Rule-related structure in population decoding with persistent influence of the irrelevant rule. A** Hypothesized templates of confusion matrices, illustrating expected patterns of similarity between rewarded trials, based on reward distinction according to the non-match rule, cue-identity rule, or both rules.
**B** Confusion matrices, averaged across rats, showing the probability of a support vector machine (SVM) decoder predicting the trial type based on neural population activity during the first and last training sessions of (**A**) and (**B**) problems under the non-match rule (confusion matrices from individual rats shown on Supplementary Fig. 5). Each matrix row shows the probability of predicting each of the possible trial types (columns), given the neural population activity for a given trial type (row).
**C** Same as (**B**), but for problems under the cue-identity rule. **D** Same as (**C**), but for the control group of rats. **E** Similarity of the averaged population decoding confusion matrices in all sessions under the non-match rule, compared to the 'Non-match template' (blue), 'Cue-identity template' (yellow), and 'both rules template' (gray, see methods). **F** Same as E, but comparing rats from the main group (circles)

and the control group (squares, dashed) in all sessions under the cue-identity rule. Blue, yellow, and gray asterisks indicate results of two-way ANOVA tests for the non-match, cue-identity, and both-rules templates, respectively. Similarity in the experimental group was significantly higher than in the control group for the non-match template ($F_{(1,35)} = 14.28$, $p = 0.0006$) and the both-rules template ($F_{(1,35)} = 5.61$, $p = 0.024$), but not for the cue-identity template ($F_{(1,35)} = 0.06$, $p = 0.811$; see Supplementary Table 9). **G** Impact of the non-match rule on population representation during cue-identity sessions, calculated as the ratio between probabilities in the main diagonal and side diagonals of the confusion matrices over all cue-identity sessions (Mann–Whitney $U$ test, $U = 1322$, $p = 0.0008$; $n = 49$ rat/session pairs in the experimental group and $n = 38$ rat/session pairs in the control group). Boxplots show medians (center lines; experimental 1.54, control 1.15), boxes the 25th–75th percentiles (experimental 1.16–2.94, control 1.03–1.49), and whiskers the most extreme non-outlier values (experimental ≈ 0.87–4.85, control ≈ 0.61–2.12), with full data ranges of 0.87–8.46 and 0.61–2.60, respectively.

underwent a shortened version of the two-rule curriculum: they first acquired the nonmatch rule on odor set A, then generalized this rule to odor set B, and finally shifted to the contradictory cue-id rule using odor sets B and then A (Fig. 8B). Prior to each session of the non-match B problem, all rats received an intraperitoneal injection of JHU37160 dihydrochloride (JH60; 0.2 mg/kg), a high-potency DREADD agonist that selectively and transiently inactivates hM4D-expressing neurons[39,40]. The use of this next-generation ligand avoids several confounds associated with earlier DREADD agonists[40,41]. Importantly, the DREADD agonist was administered only during the generalization phase of the non-match rule to odor set B, meaning OFC activity was inactivated exclusively during this stage.

Performance on the initial non-match problem (odor set A) was comparable between groups, with both hM4D and control animals reaching similar proficiency and showing no bias toward the irrelevant cue-id rule (Fig. 8C–E; $F_{(1,40)} = 1.3$, $p = 0.26$; Supplementary Table 14). However, when required to generalize the non-match rule to a new odor set under OFC inactivation, the hM4D group was significantly impaired compared to controls (Fig. 8C, D, F; $F_{(1,40)} = 31.16$, $p < 0.001$), consistent with prior work showing that OFC is critical for schema generalization[42].

The experiment then progressed to the second phase, where rats had to adapt to the new cue-id rule. At this point, OFC was no longer inactivated, since no DREADD agonist was administered prior to these sessions. Rats in the control group gradually learned to respond selectively to rewarded odors while suppressing responses based on the old non-match rule (Fig. 8D, G, yellow and blue dashed lines, respectively). By contrast, rats in the hM4D group were significantly slower to acquire the cue-id rule, reaching performance levels comparable to controls only by the final training session (Fig. 8C, G, yellow solid; $F_{(4,40)} = 5.46$, $p = 0.001$). Further, during the first two sessions, the accuracy of the hM4D group with respect to the irrelevant non-match rule was significantly lower than that of controls, suggesting they were less biased toward the prior schema ($F_{(4,40)} = 7.28$, $p < 0.001$). This impairment in acquiring the new rule persisted when the rats were switched to the cue-id A problem ($F_{(1,40)} = 31.33$, $p < 0.001$). These results are consistent with the proposal that OFC activity during schema implementation is not only necessary for successful generalization but also sets the stage for later flexibility, enabling efficient learning of a new, contradictory rule.

## Discussion

Schemas are critical to efficient behavior in the world but can also introduce problems when they become irrelevant to new environmental conditions[19,43]. Growing evidence points to a key role for the OFC in forming and deploying schemas in new situations congruent with previously acquired knowledge[19,20,44,45]. But how does this role affect the learning of a new behavioral schema that might conflict with the old one? Is representation of the prior schema in the OFC a

source of interference, slowing or disrupting learning until it is erased, or do representations in the OFC coexist or even facilitate learning of new contradictory information? Here we found evidence consistent with the latter proposal. When rats were asked to acquire a second schema that conflicted with previous learning, OFC neurons interleaved the new information with the old. This mixed representation was not observed in control rats trained only on the second schema, and it persisted even after the rats had shown expertise on new problems of the second type. Further, when encoding was examined during learning of the second, conflicting schema, clarity of representation of the prior schema was correlated with correct performance on the new one. That is, the more strongly OFC neurons represented the prior schema, interleaved with the new, the better the rats performed on the new schema. Accordingly, chemogenetic inactivation of OFC during consolidation of the initial schema impaired the rats' ability to subsequently acquire the contradictory schema. These findings indicate not only that OFC is not a source of interference when conflicting schemas must be resolved, but rather it appears to support resolution by accurately and independently representing the old rule while the new one is acquired.

Recent studies have provided insights into the neural mechanisms underlying schema formation and implementation, revealing a dynamic interaction between the hippocampus and prefrontal, especially orbitofrontal, areas in assimilating new experiences into pre-existing networks of associations[15,16,46–55]. Neural ensembles in these regions appear to converge into a hierarchical organization, structuring relationships between overlapping elements within the task space[20,45,56–60]. The assimilation of information that aligns with prior knowledge is accelerated by reactivating neuronal ensembles that represent the relevant schema, adjusting activity to incorporate novel details while preserving the low-dimensional structure relevant to common task demands[50,51,61–63]. Our findings further highlight the key role of the OFC in this process. Individual neurons in the OFC and population activity patterns across OFC ensembles dynamically adapted to quickly encode the common features of new problems of a type. Specifically, neuronal responses became more similar for stimuli associated with the same outcome, reflecting the OFC's role in organizing task-relevant information. However, as noted above, our results also suggest that the persistent representation of a prior schema in the OFC may facilitate the detection of shifts in task demands that necessitate the formation of a new schema.

Before considering the implications of these findings, it is important to consider a few caveats or limitations of the study. The chemogenetic inactivation of OFC was not selective for the specific residual representation of the prior schema, but instead broadly silenced local activity. Thus, while the impairment demonstrates that OFC contributes to this process, it cannot establish that the residual schema-related activity itself is the critical factor. Both the correlational results and the more causal inactivation findings

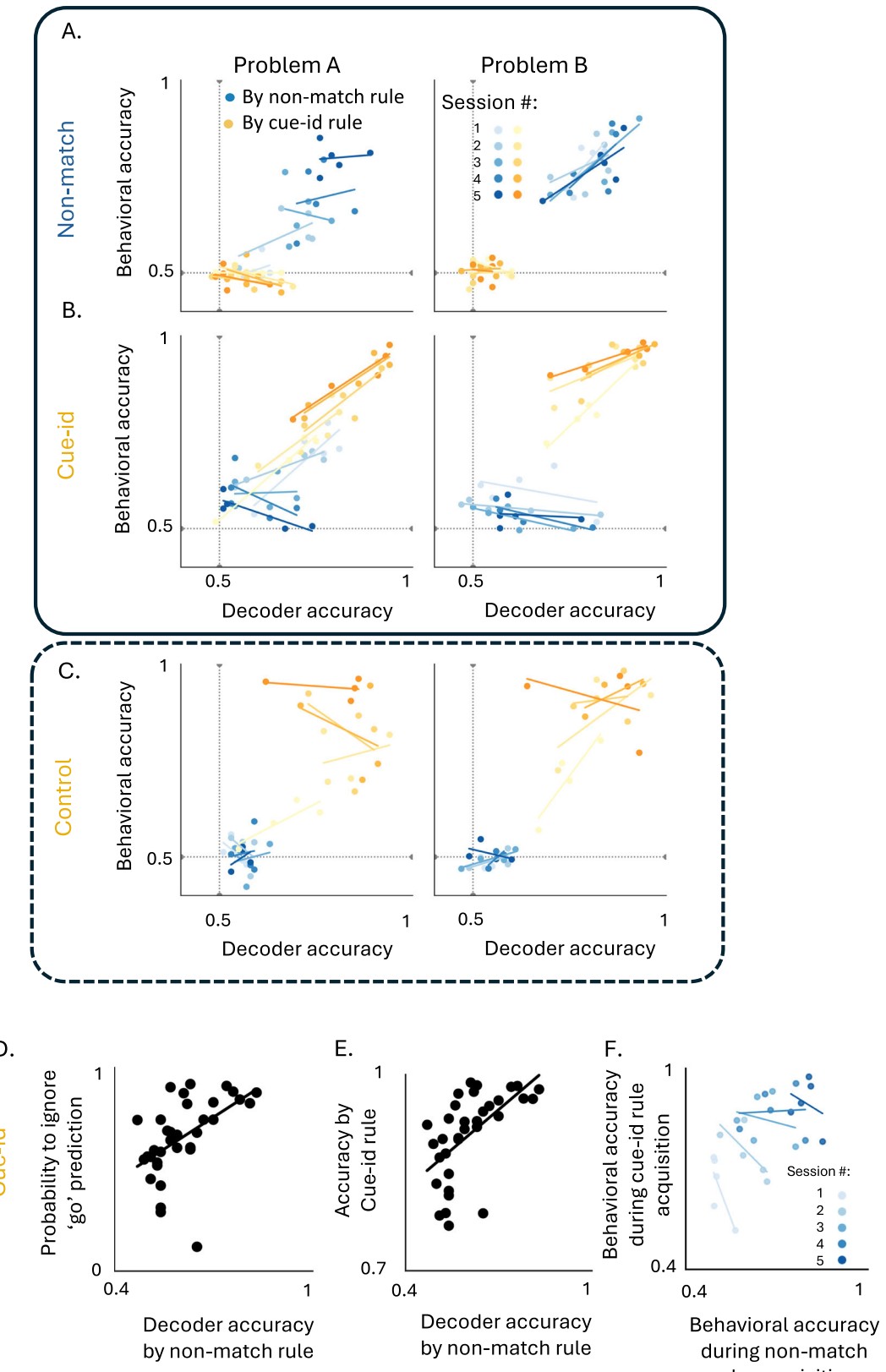

therefore leave open the possibility that other brain regions contribute to schema resolution or that different OFC signals may underlie the observed effects. Future experiments employing more targeted manipulations, like ensemble-specific inactivation or closed-loop disruption of schema-related activity[64–67], will be needed to more directly test whether the residual representation in OFC

plays a causal role in facilitating the acquisition of new, contradictory schemas.

In addition, aspects of the behavioral design constrain interpretation. In the rule-switching group, all animals learned the non-match rule before cue-identity because, in pilot work, cue-identity–first training was substantially slower and more variable,

**Fig. 7 | Effects of residual encoding on behavior during learning of orthogonal rules. A** Correlation between behavioral accuracy and population decoder accuracy according to the non-match rule (blue) and cue-identity rule (yellow) in Problem (**A**) (left) and (**B**) (right) during the non-match training phase. Classifiers were trained separately for each rat and session. Decoder accuracy was significantly correlated with behavioral accuracy according to the non-match rule in both Problem (**A**) (simple linear regression: $R^2 = 0.57$, $\beta = 0.97$, $p = 0.0001$) and (**B**) ($R^2 = 0.50$, $\beta = 0.75$, $p = 0.0008$). No significant correlation was observed for the cue-identity rule (Problem (**A**): $R^2 = 0.12$, $\beta = -0.13$, $p = 0.086$; Problem (**B**): $R^2 \approx 0$, $\beta = 0.02$, $p = 0.90$). Full regression and interaction models are shown in Supplementary Table 10. **B** Same as (**A**), but during the cue-identity training phase. Decoder–behavior correlations remained strongly positive for the cue-identity rule in both Problem (**A**) ($R^2 = 0.88$, $\beta = 0.92$, $p = 2.2 \times 10^{-12}$) and Problem (**B**) ($R^2 = 0.56$, $\beta = 0.61$, $p = 0.0003$). Correlations with the irrelevant non-match rule reversed in later sessions of Problem (**A**) ($R^2 = 0.23$, $p = 0.016$). Session-level slope analysis confirmed this reversal: early sessions showed positive slopes (0.89, 0.37; $r = 0.86$, 0.95), whereas later sessions flattened or inverted (0.03, −0.46, −0.34; r = 0.06,

−0.58, −0.87). Early slopes were positive on average (0.63) and late slopes negative (−0.26), with a significant Fisher Z difference ($z = 4.20$, $p = 2.7 \times 10^{-5}$; Supplementary Table 11). **C** Same as (**A**–**B**), for the controls. No significant decoder–behavior relationship was observed under the non-match rule (Problem (**A**): $R^2 = 0.002$, $p = 0.84$; Problem (**B**): $R^2 = 0.10$, $p = 0.20$). For the cue-identity rule, no correlation appeared in Problem (**A**) ($R^2 = 0.13$, $p = 0.12$), but a positive correlation was found in Problem (**B**) ($R^2 = 0.26$, $\beta = 0.62$, $p = 0.033$). **D** Correlation between irrelevant-rule decoder accuracy and the proportion of trials labeled "go" by this decoder but ignored by the animal ($R^2 = 0.27$, $\beta = 1.14$, $p = 0.002$; Supplementary Table 12). **E** Correlation between irrelevant-rule decoder accuracy and behavioral accuracy for the relevant cue-identity rule ($R^2 = 0.34$, $\beta = 0.40$, $p < 0.001$; Supplementary Table 12). **F** Correlation between cue-identity behavioral accuracy during its acquisition and non-match behavioral accuracy during its initial learning. No significant relationships were observed (Pearson $p > 0.1$; Supplementary Table 13), indicating that cue-identity learning speed was not predicted by earlier non-match performance.

making groups impractical and poorly comparable. Consequently, sequence-dependent effects cannot be excluded. Counterbalanced training in future studies would strengthen inferences about OFC's role in facilitating schema-based learning of conflicting rule. A related concern is that decoding contrasts may reflect reward contingency rather than schema representation, as rewarded and non-rewarded trials differ along both axes. However, all decoding analyses were restricted to the odor-sampling period prior to reward delivery, isolating neural activity related to rule representation. Moreover, the direction of the effect argues against a simple reward code: stronger representation of the old rule predicted a greater tendency to disregard its reward contingencies and act according to the new rule. Thus, rather than reflecting a trivial encoding of reward outcome, the persistence of the old schema representation in OFC appears to predict its selective gating when incongruent with current task demands.

Caveats aside, a potential role for the OFC in facilitating the learning of conflicting information is reminiscent of the long-recognized importance of the OFC to so-called cognitive flexibility[68,69], epitomized by reversal learning. The OFC has been found to be necessary for reversal learning in many, though not all, settings[24,70–76]. The current results provide a more specific basis for this involvement in the accurate representation of the prior rule, independent of the new, so that errors or mistakes can be effectively signaled even while the new rule is acquired. Such a role should be particularly important when changes occur against a background of strong priors, since it is under these conditions, illustrated by the over-training reversal effect[37,38], that representation of the prior information would be most effective. Accordingly, the OFC is most critical for initial reversal learning, and for reversal learning when contingencies have been stable for long periods[70,71]. When contingencies are rapidly changing, the OFC is unnecessary for accurately tracking the best choice, and it has even been shown to hinder reversal under these conditions[71,75,77,78].

The current results also provide a novel explanation for the involvement of the OFC in cognitive flexibility in the persistent multiplexing of the schemas—the generalized cognitive maps[79]—deployed to guide behavior. This multiplexing is superior to forgetting in several ways. Most obviously, multiplexing ensures prior knowledge remains accessible for future transfer, allowing flexible adaptation or even spontaneous recovery if new problems fitting the old schema are encountered[80]. However, in addition to this valuable function, multiplexing the old schema independently from the new also makes it possible for the old information to support stronger error signaling to drive learning, even as the new information is laid down, whereas if the two rules were confused or the old was eliminated, teaching signals would weaken more quickly. In this light, it is

interesting to consider that the OFC is not generally necessary for established performance. OFC manipulations have been shown repeatedly not to impact task performance after the relevant relationships and rules have been learnt; this includes settings such as economic choice, in which core functions of the OFC are purported to be at risk[81,82]. Instead, the OFC appears most necessary during learning or when new information must be acquired and used to change behavior[33,83–88]. This suggests that the OFC is typically functioning as a follower in using existing information, and its role becomes decisive when integrating new and conflicting information which is required for normal behavior. Viewed from this perspective, the ability to hold two independent sets of information online may be particularly important for comparison and to redirect the learning toward the relevant information. Consistent with this, the OFC has broad influences on downstream areas, both for supporting the representation of associative information in subcortical regions like amygdala and striatum[89–95], while also contributing to error signaling by midbrain dopamine neurons[96–101]. If the multiplexed information evident here in OFC ensembles can be demixed through selective projections or downstream filtering, the OFC would be in a powerful position to simultaneously serve both roles[102,103].

The finding that OFC maintains a latent trace of prior learning, even when it conflicts with current goals, offers a concrete, biologically grounded strategy to mitigate practical issues that arise in deep neural network architectures used for artificial intelligence (AI), where new learning typically overwrites prior knowledge, a phenomenon known as catastrophic forgetting[104]. Such architectures allow an AI agent trained to play chess, where controlling the center or maintaining piece activity are crucial to success, to more quickly shift to the game of Go, where early play favors corners and sides over the center of the board. While approaches such as replay buffers, regularization, or mixture-of-experts architectures have been proposed to allow retention of the prior chess principles in this situation, these solutions are computationally demanding and lack native, context-dependent suppression[105]. Indeed, even without such added overhead, AlphaGo Zero, a landmark system achieving superhuman performances in Go, required massive self-play and computing, consuming energy orders of magnitude above the human brain's modest -20-W power budget[106–108]. Our findings suggest that OFC retains prior rule representations while supporting new learning, much like a human chess player who can benefit from, yet adaptively suppress, prior schemas when entering a game of Go. Understanding the underlying neural mechanisms that make this possible could inspire more efficient strategies for knowledge transfer in AI, enabling more general use systems able to leverage rather than erase prior knowledge when faced with new rules or environments[109].

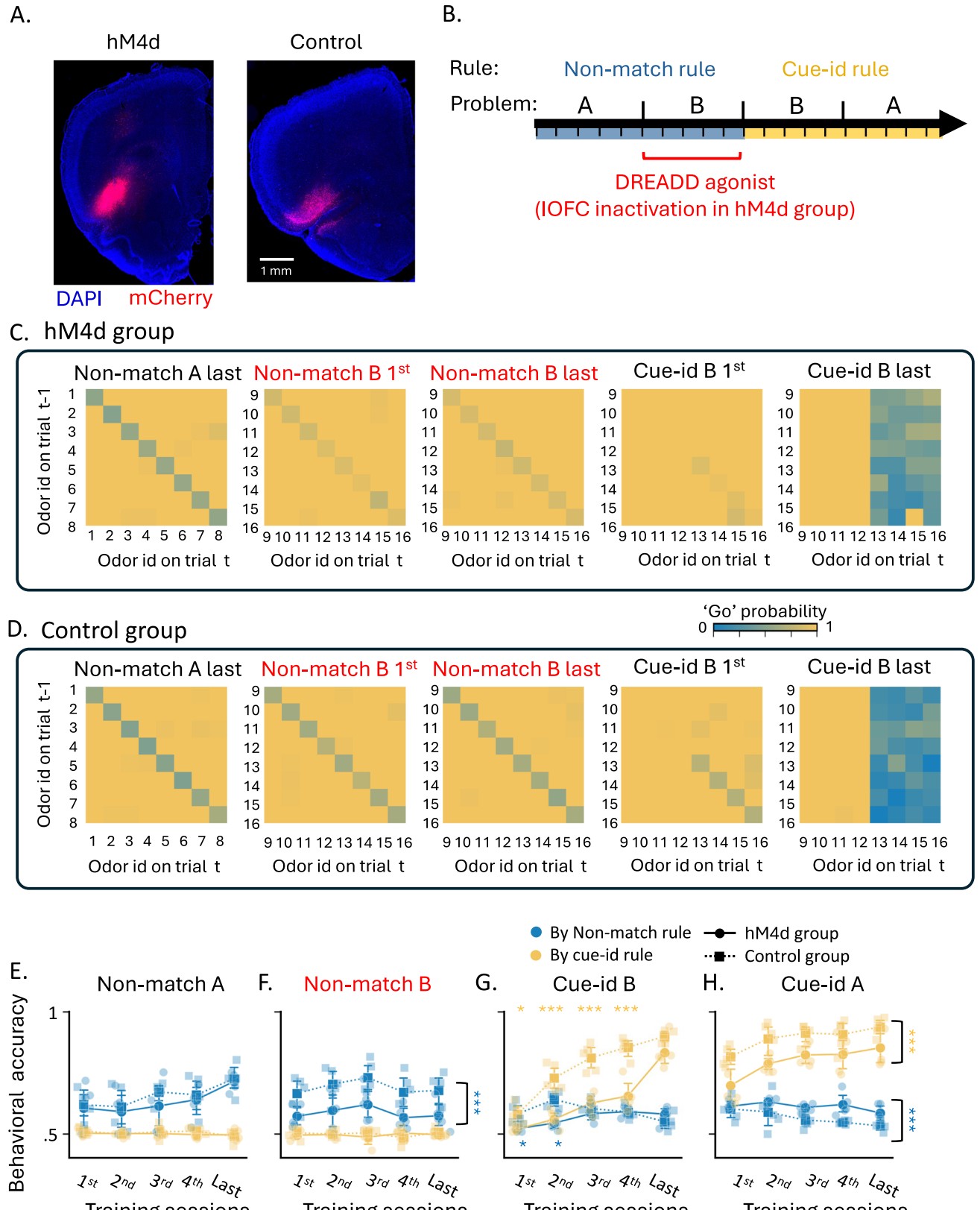

## Methods

### Experimental model and subjects

The recording study used nine male Long-Evans rats (Charles River), weighing between 300 and 400 g and approximately 4 months old at the start of training ('rule-switching' group: $n = 5$, 'control' group: $n = 4$). An additional cohort of ten Long-Evans rats (six males, four females) was used in the complementary chemogenetic inactivation experiment. The rats were housed individually under a 12-h light/dark cycle (lights on at 7:00 pm) at the AAALAC-accredited animal facility of the National Institute on Drug Abuse Intramural Research Program (NIDA-IRP), with unrestricted access to food. Housing rooms were maintained at a stable ambient temperature of 20–22 °C and relative

**Fig. 8 | Effects of OFC inactivation on the learning of orthogonal rules.**
**A** Representative photomicrographs of viral transfection in hM4d and control rats. comparable viral expression was observed in all animals (hM4D: $n = 5$; control: $n = 5$). **B** Curriculum overview of the inactivation experiment. DREADD agonist was administered only before sessions in the Non-match B problem. **C** Probability of responding 'go' across all trial types (different odors in match/non-match configurations) for hM4d group. **D** Same as (**C**) but for control group. **E** Accuracy in problem non-match A under the non-match (blue) and cue-identity (yellow) rules (data are shown as mean ± SEM with individual data points overlaid). Solid lines represent the hM4d group ($n = 5$ rats), while dashed lines represent the control group ($n = 5$ rats). Non-match accuracy increased with training (main effect of Session in two-way ANOVA: $F(4,40) = 3.79$, $p = 0.011$). Cue-identity accuracy showed no significant effects (see Supplementary Table 14). **F** Same as (**E**), but for problem

non-match B (red title indicates this is the only problem when DREADD agonist was administered). Controls maintained higher non-match accuracy than the hM4d group (main effect of Group in two-way ANOVA: $F(1,40) = 31.16$, $p < 0.001$), with no training-related change. Cue-identity accuracy showed no significant effects. **G** Same as (**E**), but for cue-identity B problem. Non-match accuracy decreased with training and was lower in the hM4d group, with a significant Session × Group interaction. Cue-identity accuracy increased with training, with lower accuracy in the hM4d group and a significant Session × Group interaction. **H** Same as (**E**), but for the cue-identity A problem. Non-match accuracy decreased with training and was lower in the hM4d group. In contrast, cue-identity accuracy increased with training and was higher in the control group (all ANOVA results are in Supplementary Table 14).

humidity of 40–60%, consistent with NIH animal care standards. Water was removed the day before testing sessions, and the rats were allowed 10 min of water access in their home cages after each testing session. On days without testing, the rats had free access to water. All procedures were carried out in compliance with the guidelines of the U.S. National Institutes of Health (NIH) and were approved by the Animal Care and Use Committee (ACUC) of NIDA-IRP.

### Stereotaxic electrode implantation
Rats were implanted with 2-4 drivable electrode bundles, each containing 16 nickel-chromium wires (25 μm diameter, AM Systems, WA), totaling 32-64 electrodes, targeting the lateral orbitofrontal cortex (lOFC) at coordinates AP: 3 mm and ML: 3.2 mm. The electrode bundles were embedded in 27-gauge stainless-steel hypodermic tubing and mounted in a custom-built, 3D-printed microdrive. Before surgery, the bundles were trimmed to 1-2 mm with fine bone-cutting scissors (Fine Science Tools, CA) and spaced to maintain at least 25 μm separation between wires. During surgery, rats were anesthetized with isoflurane (3% induction, 1–2% maintenance in 2 L/min O2) and secured in a stereotaxic frame (Kopf Instruments, Tujunga, CA) for electrode implantation. Electrode tips were initially positioned 4.2 mm ventral to the brain surface. Headcaps were affixed using 0–80 1/8" machine screws and dental acrylic, then encased in a custom 3D-printed protective cover. Post-surgery, rats received Cephalexin (15 mg/kg) orally twice daily for two weeks to prevent infection.

### Stereotaxic viral injection
Rats were anesthetized with 1–2% isoflurane and received bilateral injections of either AAV8-CaMKIIα-hM4D(Gi)-mCherry (a Gi-coupled designer receptor exclusively activated by designer drugs, DREADD) or AAV8-hSyn-mCherry (control), both obtained from Addgene (Cambridge, MA). Injections targeted the lateral OFC (coordinates: AP – 3.0 mm, ML ± 3.2 mm, DV – 4.4 and −4.5 mm from the brain surface). At each site, 0.5 μL of viral solution was delivered at a rate of 0.1 μL/min using an infusion pump.

### Odor-guided discrimination tasks
Behavioral training was conducted in aluminum boxes (∼18 inches per side) equipped with an odor delivery port and a sucrose solution well. Task execution was controlled by a custom C ++ program controlling relays and solenoid valves, with infrared sensors detecting entries into the odor and fluid ports. Each trial began when two house lights were illuminated above the odor port, prompting the rat to nosepoke within 5 s. Upon entry, a 500 ms delay was followed by odor presentation, during which the rat was required to remain in the port for an additional 300 ms; early withdrawal aborted the trial. After this period, rats could exit the port, stop odor delivery, and had 2 s to make a response at the fluid well. For rewarded trials, a response triggered a 50 μL sucrose solution (5% w/v) delivery after a 1000 ms delay. If no response occurred or the trial was non-rewarded, the house lights turned off, initiating a 4-second inter-trial interval.

Before odor training, rats were shaped to nosepoke and respond at the well for a reward. After 3-4 shaping sessions, they were trained on a series of odor-guided discrimination problems involving two sets of 8 unique odors (A and B), each set predicting reward based on one of two orthogonal rules (Fig. 1a). The first rule was a non-match rule, under which a response was rewarded if the odor presented on the current trial differed from that presented on the previous trial. Rats were trained for at least 5 sessions on each problem before advancing to a new problem with the same rule but different odors. In the second phase, the task rule changed to a cue-id rule, under which the match comparison became irrelevant and rewards were contingent only on the identity of the odor presented on the current trial. Here, half of the odors (1–4/9–12) were associated with a potential reward ('go' odors), while the remaining odors (5–8/13–16) were not ('no-go' odors).

### Inactivation experiment
Prior to each non-match B session, all rats received an intraperitoneal injection of JHU37160 (0.2 mg/kg in 0.9% NaCl), after which they were returned to their home cage for at least 15 min before the start of the session to allow the DREADD agonist to effectively inhibit transfected lOFC neurons in the hM4D group.

### Single-unit electrophysiology
Neural recordings were obtained using the Plexon OmniPlex system (v2.7.0; Plexon, Dallas, TX). Neural signals were digitized, amplified, and bandpass filtered (250–8000 Hz) to isolate spike activity. Thresholds were manually set on each channel to capture unsorted spikes. Behavioral timestamps were synchronized with neural data in real time. After recording, spikes were sorted offline using Offline Sorter (v4.0; Plexon, Dallas, TX). Single units were isolated in 2D feature space (PC1, PC2, nonlinear energy), after which unit and event timestamps were exported to Matlab for further analysis. The sorted data were exported for analysis in MATLAB (2021a; MathWorks). Electrodes were advanced approximately 120 μm between odor discrimination tasks to sample new neuronal populations, although neuron identity across sessions was not assumed.

### Statistics and reproducibility
All analyses were conducted in Python 3.9 using DataSpell (JetBrains, 2024). Error bars in figures indicate the standard error of the mean (SEM). Behavioral performance and comparisons with control groups were evaluated using two-tailed unpaired t tests for single-factor analyses, or two-way repeated-measures ANOVAs for multi-factor designs, followed by Tukey's post hoc tests where appropriate. Linear regression was used to assess correlations between variables. Statistical significance was defined as p < 0.05 for all tests. To provide full transparency, detailed outputs of all statistical tests are reported in the Supplementary Tables. In addition to parametric tests, permutation analyses (1000 iterations) were used to generate null distributions for classifier accuracy and other model-based measures. This non-parametric approach makes no assumptions about the underlying

data distribution and is therefore well suited for neural population analyses, where independence across trials and normality of residuals may not be guaranteed. Sample sizes were not predetermined by formal power analysis but were consistent with standards in the field. Group allocation and counterbalancing were performed pseudo-randomly. Data were assumed to follow a normal distribution, though this assumption was not formally tested.

## Behavior analysis

Behavioral data were collected using custom software written in C++, which sent event timestamps to the electrophysiological recording system. Raw data were processed in MATLAB 2021a (Mathworks, Natick, MA) to extract the time spent in the odor and reward ports relative to trial initiation. Further analyses were performed using Python 3.9 and Jupyter notebook. Behavioral accuracy was quantified by the percent of trials on which the rats responded correctly according to the non-match rule or the cue-id rule. Number of trials to reach learning criterion was calculated as the number of trials in each session it took to cross 80% correct across 30 trials. Group differences were assessed with statsmodel library by ANOVA, with significant results followed by Tukey's Honest Significant Difference post hoc test for pairwise comparisons at an alpha level of 0.05, with corrections for multiple comparisons applied where appropriate.

## Single unit analysis

The spike train for each isolated single unit was aligned to the decision time (time of un-poking from the odor-port). Spike number was counted with a bin of 50 ms. A peri-stimulus time histogram (PSTH) was generated by calculating the mean neural response across different trials. For each trial type, the mean response was smoothed using a convolution of a moving average filter, defined as a uniform filter with a window size of 5-time bins.

For each single unit, z-scores were calculated separately for rewarded and non-rewarded trials by normalizing the mean firing rate in the response window (500 ms prior to odor un-poke) against baseline activity (1 second prior to trial initiation). The z-score for non-rewarded trials was then subtracted from that of rewarded trials to obtain the delta z-score, representing the differential standardized activity between trial potential outcomes. Units were subsequently ordered by their latency to peak activity, and the colormap was trimmed between −2 and 2 to enhance visibility. To assess rule-specific neural modulation, each single unit's delta z-score was computed separately for trials based on the two underlying rules.

To compare rule-dependent selectivity across the population, we applied the parameter-free ZETA test to identify neurons with significantly different firing rates in the window between poke and un-poke from the odor port for rewarded versus non-rewarded trials[35]. Briefly, the neuronal activity in response to the odor sampling period was compared against a shuffled null distribution created by permuting the activity across trials. This algorithm identifies responsive time windows without predefined parameters, ensuring that only statistically significant deviations from the null model are classified as true responses. Neurons that exhibited a significant evoked response and differential firing across these two conditions were labeled as selective for that rule. This process was conducted separately for the non-match and cue-id rules.

## Population analysis

To explore and quantify the representation of the learned rules by the neural population activity, we employed Uniform Manifold Approximation and Projection (UMAP), a non-linear dimensionality reduction technique, to embed neural activity into a low-dimensional space[110,111]. Neural firing rates were extracted from the 500 ms period preceding the un-poking from the odor port and normalized using a standard scaling method to ensure consistency across datasets. Recordings from equivalent training sessions of different rats were normalized independently, and the data were subsequently aligned by trial type to generate pseudo-ensembles. UMAP, implemented using the Python library umap-learn, was then applied to the aligned dataset, resulting in three-dimensional embeddings. These embeddings were visualized for each trial, with data points distinguished by odor identity (color) and trial configuration (marker style). To further quantify the relationships between the neural representation of different trial types in the UMAP space, we calculated the distances between their centroids and visualized these relationships using a dendrogram. For each trial type, the mean embedding was computed by averaging the UMAP dimensions for all data points within that trial type. The distances between these centroids were then calculated using a normalized Euclidean distance metric, which accounted for variance within each trial type. Hierarchical clustering was performed using the Ward linkage method, implemented in the Python library scipy, to generate a linkage matrix, which was subsequently used to construct a dendrogram. This dendrogram provided a hierarchical visualization of the relationships and separability of task-specific neural activity patterns in the UMAP space.

Additionally, the distances between the representations of odors in the match versus non-match configurations and the distances between odors across the rewarded versus non-rewarded groups were calculated for each session. To evaluate the robustness of these representations, we applied a Leave-One-Subject-Out (LOSO) sensitivity analysis: for each session, embeddings and centroid distances were recomputed repeatedly while systematically excluding one rat at a time. The mean distance across LOSO iterations was then taken as the session-level value. To assess statistical significance, LOSO-derived means were compared against a permutation-based null distribution generated by shuffling trial labels within each session (1000 shuffles). Two-tailed p-values were computed as the proportion of permuted distances greater than or equal to the observed LOSO mean, and multiple comparisons across sessions were corrected using the Holm–Bonferroni method.

For group-level comparisons, for each session, embeddings and centroid distances were recomputed while systematically excluding one rat at a time, and the resulting distances were averaged across LOSO iterations to obtain group-level means. We then computed the observed difference in these LOSO-averaged means between the experimental and control groups. To assess statistical significance, this observed difference was compared against a null distribution generated by calculating the difference between the corresponding permutation-based null distributions from each group (1000 shuffles per session).

To further assess the geometry of the neural representation, pseudo-ensembles (aligned by trial type and concatenated across neurons) were used to compute cross-condition generalization performance (CCGP[36]) separately for non-match and cue-identity. Here, conditions are the different odor identities within the same contingency. For each rule, a linear classifier decoded rewarded vs. non-rewarded trials using cross-condition splits: the model was trained on a subset of odors and tested on held-out odors from that same contingency, repeated over 500 random partitions. Session-level performance was summarized as the mean and percentile-based 95% CI across partitions; 0.5 was treated as chance. Significance versus chance was assessed nonparametrically from the empirical accuracy distribution (two-tailed tail probability).

To quantify representational geometry, we estimated a linear decision axis for each rule and computed their cosine similarity from normalized weight vectors. For each rule, we trained a logistic regression classifier to distinguish trial types (rewarded vs. non-rewarded) based on the neural population activity vectors. To ensure balanced training, trials were randomly subsampled within each bootstrap iteration to achieve equal class sizes. This process was

repeated 1000 times to generate a bootstrap distribution of similarity values. In each iteration, the classifier weights obtained for the non-match ($w_{NM}$) and cue-identity ($w_{CI}$) rules were projected into the original feature space, and the cosine similarity between them was calculated as

$$\cos(\theta) = \frac{w_{NM} \cdot w_{CI}}{\| w_{NM} \| \| w_{CI} \|}$$

This metric captures the alignment between the linear decision axes associated with each rule (0 = orthogonal, 1 = aligned, −1 = anti-aligned). The resulting distribution of cosine values across bootstraps was used for group-level statistical analysis.

To assess the alignment of the neural population activity with each rule, we employed a support vector machine (SVM) classifier from scikit-learn library. The SVM model was trained separately for each animal and training session to predict trial type based on neural responses during the odor sampling period. A leave-one-out cross-validation strategy was applied to generate trial-wise prediction probabilities, which were visualized through confusion matrices. These matrices where then compared against the template matrices to calculate similarity to each template. To test the significance of similarity between the neural confusion matrix and rule-based templates, against a null hypothesis of the similarity to each template happening by chance, we conducted a permutation test (1000 permutations). For each permutation, we randomly shuffled rows and columns of the neural confusion matrix to create a permuted matrix, then computed its similarity score with each target template by applying a scoring matrix. After calculating actual similarity scores for the unshuffled matrix, we compared these to the permutation-based scores to obtain p-values, indicating the proportion of permuted scores meeting or exceeding the observed score. This non-parametric test evaluated the alignment between neural activity and each rule template.

To further assess the decodability of each rule from the neural population of each rat, we trained additional classifiers to differentiate between rewarded and non-rewarded trials based on each rule. Using 1000 permutations, we performed a permutation test to evaluate classifier accuracy as the mean of cross-validated scores and calculated an empirical p-value for statistical significance. Decoder fidelities reported in the main text (Fig. 7) and Supplementary Fig. 6 therefore represent per-session, per-animal values rather than pooled analyses across animals or sessions.

To isolate rule representation from the potential influence of animal choice, we regressed out choice-related variance from the neural data. This was achieved by using a linear regression model, where animal choice (treated as a predictor variable) was fitted to the neural response data. The residuals, representing neural activity after accounting for choice, were then used for subsequent SVM decoding analysis. This approach ensured that the classification accuracy reflected rule-specific neural patterns independent of the animal's choice behavior.

The probability of ignoring the 'go' prediction was calculated as the proportion of trials that were labeled as rewarded trials by the 'non-match' classifier, but had not resulted in a 'go' action.

## Reporting summary
Further information on research design is available in the Nature Portfolio Reporting Summary linked to this article.

## Data availability
The data described in this study available on Figshare at: https://figshare.com/s/c44a2d3fc5d00b5116a1 Additionally Source data are provided with this paper.

## Code availability
Custom analysis code available on GitHub at: https://github.com/IdoMaor/ofc_schemas https://doi.org/10.5281/zenodo.17945155

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

## Acknowledgements

The authors thank our current and former laboratory members, for valuable discussions and technical assistance and to the animal care and veterinary staff for their support. This research was supported by the Intramural Research Program of the National Institutes of Health (ZIA-DA000587 to G.S.) and by a fellowship from the NIH Center on Compulsive Behaviors (CCB Fellowship to I.M.). The contributions of the NIH

authors are considered Works of the United States Government. The findings and conclusions presented in this paper are those of the authors and do not necessarily reflect the views of the NIH or the U.S. Department of Health and Human Services.

## Author contributions

I.M. designed the study, performed experiments, analyzed data, and wrote the manuscript. J.A. and I.A. conducted behavioral and electrophysiological experiments. Y.Z. and F.P. contributed to data analysis and interpretation. Y.K.T. and E.H. assisted with experimental design and methodological planning. G.S. supervised the project and contributed to study design and manuscript preparation.

## Funding

## Competing interests

The authors declare no competing interests.
