## [Transparent Peer Review file · Nature Communications]

Persistent representation of a prior schema in the orbitofrontal cortex facilitates learning of a conflicting schema

Corresponding Author: Dr Geoffrey Schoenbaum

Version 0:

Reviewer comments:

Reviewer #1

(Remarks to the Author)

This work investigates how the orbitofrontal cortex (OFC) contributes to learning new behavioral rules that conflict with prior knowledge. Using a well-designed odor-guided decision task in rats, combined with single-unit electrophysiology and population-level decoding methods, the authors report that OFC neural activity retains representations of previously learned schemas even after a new rule is adopted. Interestingly, the strength of this residual encoding is positively correlated with performance on the new rule, suggesting a facilitative rather than interfering role.

This is a timely and conceptually important study that addresses the neural basis of schema-dependent learning and cognitive flexibility. However, some causal inferences may not be fully supported by the data, and some methodological clarifications are needed to support key claims as outlined below.

Major Issues

First, the central claim that residual representations of the prior schema facilitate learning of a new rule is stated in causal terms but rests entirely on correlational evidence. Although a positive relationship between decoder fidelity for the prior rule and behavioral performance on the new rule is reported, there is no experimental manipulation to demonstrate a mechanistic role. It remains possible that both effects arise from a shared third factor, such as general task engagement or learning ability. Without causal disruption (e.g., OFC inactivation or schema-specific interference – note that I am not necessarily calling for more experiments), the authors should temper the interpretation of their results and clarify that the facilitative effect is an inference, not a demonstrated mechanism. The control group (rats trained only on the cue-identity rule) is certainly helpful in showing that residual non-match encoding depends on prior experience but it still does not directly test whether this residual representation is necessary for accelerated learning. The authors should more clearly acknowledge this limitation when interpreting the putative facilitative role of prior schema encoding. This issue is also more prominent because of the relatively small sample size (eight male rats) and the lack of detailed reporting on how many neurons were recorded per rat, per session, and per condition. Given the variability in both behavior and neural encoding, it is essential to show that the reported effects are robust across subjects. Cross-animal reliability measures or bootstrapping across animals would help to substantiate the population-level claims. Was there a reason to not include female rats in the study?

Second, for the neural decoding methods, greater transparency and methodological detail may be warranted. It is not clearly stated whether classifiers were validated within sessions, across sessions, or across subjects. Similarly, the use of pseudo-populations across animals raises concerns about overfitting and the independence of the training and test data. The authors should clarify how they ensured that the decoding results reflect meaningful neural structure and not artifacts of trial averaging or classifier bias. Additionally, the potential confound between reward contingency and schema representation deserves further consideration, as most decoding contrasts involve rewarded vs. non-rewarded trials that differ along both axes.

A final concern relates to the interpretation of "multiplexing" in OFC. The idea that OFC simultaneously encodes multiple schema representations is interesting but underspecified. The authors do not provide a clear analysis of whether the same neurons encode both schemas, or whether distinct populations are involved. This distinction is important, both mechanistically and conceptually. Dimensionality reduction or clustering analyses could help clarify whether the old and

new rule representations occupy separable neural subspaces or overlap within a shared coding space.

Minor Issues

Several statistical details are not fully described. For example, the authors report many ANOVAs and permutation tests across figures and sessions, but it is not always clear whether corrections for multiple comparisons were applied. Exact p-values, effect sizes, and confidence intervals should be reported consistently, especially for the key correlations between neural decoder accuracy and behavior.

The manuscript is generally well written, though there are instances where the phrasing overstates the results (e.g., "culminating in psychopathology" in the abstract) or introduces ambiguity. Some figure panels are densely labeled, and the font sizes in certain plots are small. Improvements to figure readability (particularly in the decoding and UMAP plots) would enhance accessibility.

The concluding discussion section includes a thoughtful comparison between biological and artificial learning systems, which is intellectually interesting but somewhat tangential. This comparison could be trimmed or better integrated into the main themes of the manuscript.

Reviewer #2

(Remarks to the Author)

Maor and colleagues use an odor-discrimination task to explore how the orbitofrontal cortex (OFC) represents two competing behavioral rules. Rats were first trained to follow a "non-match" rule (respond when the current odor is different from the previous one) and later learned a "cue identity" rule (respond only to four specific odors, regardless of sequence). Recordings from the lateral OFC showed that both behavior and neural activity adapted with repeated exposures to the task, suggesting schema formation, and after switching rules, most neurons adjusted to the new schema, though a small group continued to encode the old rule. Interestingly, the stronger this persistent activity was, the faster the rats learned the new rule.

The study uses a creative task design and a broad range of analytical tools to explore the data. This work significantly advances our understanding of how the brain supports rule-based decision-making and introduces compelling new insights. However, I have several concerns, primarily regarding methodological clarity and analytical rigor. These are listed in order of importance

1. The correlation results in Figure 7 are based on only four data points. While I appreciate the technical and logistical challenges of in vivo electrophysiology, this limited sample raises concerns about the robustness of the findings. I suggest either removing this result or including additional animals to strengthen statistical validity. If this is not possible, these limitations should be explicitly discussed and acknowledged clearly in both in the manuscript and abstract.

2. It is unclear whether the analyses included only correct trials. If so, the authors should explicitly detail how trial variability was addressed, particularly concerning trial stratification for analyses like SVM decoding. Differences in trial counts at different training stages could significantly bias decoding results. Conversely, if incorrect trials were included, the authors must clarify how differences in trial correctness between experimental and control groups were accounted for, especially given early discrepancies in 'correct' trial proportions for non-match rule between these groups during initial cue training.

3. For results shown in Figure 5e, the statistical comparison should be directly conducted between control and experimental groups rather than indirectly through comparisons to shuffle conditions. The authors could, for instance, test whether the difference between experimental and control conditions is significantly greater than differences observed between two null distributions.

4. The analysis presented in Figure 6g lacks clarity regarding the experimental phase from which data were derived. Again, given that early cue identity training produces notable differences in correct trial proportions between groups (experimental and control), the presented analysis could potentially be biased by these initial performance disparities.

5. The authors employ a mix of parametric, non-parametric, and permutation statistical methods. It would be beneficial to explicitly justify the rationale behind selecting these various statistical approaches throughout the manuscript.

6. The text results associated with Figure 6f is unclear, as this panel contains two separate plots. The authors should indicate whether the reported statistical results pertain to both plots or only one.

7. Figure 3C Adding session numbers to the x-axis of Figure 3c would improve plot readability and facilitate interpretation.

Reviewer #3

(Remarks to the Author)

This manuscript presents exciting and novel findings demonstrating that orbitofrontal cortex (OFC) neural representations track rule learning and, as hypothesized, both OFC activity and task performance were facilitated by prior learning of a different rule. A particularly important and unexpected result was that OFC ensembles continued to represent the previously

learned rule even after rats acquired the new one—and more strikingly, the stronger the OFC representation of the old cue, the better the performance after the rule switch. These findings provide compelling evidence that persistent encoding of prior schemas in the OFC may actively support, rather than hinder, behavioral flexibility. The results will be of broad interest to neuroscientists studying cognition, learning, and prefrontal function. While the manuscript is strong overall, there were some minor issues with clarity, including a somewhat niche chess analogy, a lack of explanation for the fixed rule order, and the need for clearer figure presentation and labeling—particularly in Figures 2I and 3B/C. Moreover, one limitation was that only male rats without justification.

The work will be of significance to the field because it advances our knowledge of how OFC neural ensembles represent new and old information, adding empirical evidence to the research on schemas. Some additional literature review that mentions the work on schemas and memory consolidation in the hippocampus could be relevant here even though the manuscript focuses on OFC and rule switching. See <https://doi.org/10.1126/science.1135935>

The conclusion, that the OFC contributes to new learning by representing the old rule as the new rule is learned, is strongly supported by the data.

Only minor issues are noted with the clarity of the presentation. These are:

- The chess analogy could be made more general. Some of the specifics will only be understandable to readers who are familiar with chess strategy.
- The authors should explain why they did not counterbalance the order of the rules. All rats learned the nonmatch rule first, followed by the cue-id rule. Was there some practical reason for this? What would happen if rats learned the cue-id rule first, then the nonmatch rule. Some of this concern is mitigated by the control group that learned only the cue-id rule. I am not suggesting that the study design was flawed. Instead, I would like the authors to explain why they didn't employ this approach which is very common in the field of behavioral neuroscience.
- For Fig. 2, Panel I, the bars and horizontal lines did not come through on the pdf, making the figure legend and caption confusing.
- Fig. 3B: More details are needed. How were the units ordered in the z-score grayscale map?
- Fig. 3C: Please clarify what is being shown in the rightmost plot. The title and legend are the same as the leftmost plot and the caption does not give any clues.

The methodology is sound, except there is no justification for using only male rats. Single-sex (male-only) studies are becoming increasingly rare in science, especially with no justification included. In fact, most top tier journals as well as federal funding agencies are requiring not only inclusion of both sexes, but that sex be analyzed as a variable.

There is enough detail included in the methods for the work to be reproduced.

Reviewer #4

(Remarks to the Author)

This manuscript by Maor et al. investigates the role of the orbitofrontal cortex (OFC) in supporting flexible learning when confronted with conflicting schemas. Through a combination of behavioral experiments, single-unit neural recordings, and advanced population decoding analyses, the study reveals that the OFC retains representations of prior schemas while concurrently encoding new ones. This dual representation provides a framework for flexible adaptation to environmental demands and highlights how the brain resolves conflicts between old and new behavioral rules.

The core finding is that prior schemas are not overwritten but persist in the OFC even during the acquisition of conflicting schemas. This challenges traditional theories of interference and forgetting and offers an alternative neural mechanism underlying cognitive flexibility.

I like the paper a lot in general, but with one major concern. While the authors convincingly demonstrate that animals with stronger representations of prior schemas perform better on new tasks, it is not entirely clear whether this relationship is causal. An alternative explanation could be that these animals are simply better learners overall, independent of the persistence of the old schema. Addressing this potential confounder would strengthen the authors' interpretation. For instance, maybe the authors could look at the baseline learning ability or conduct experiments to selectively manipulate the representation of the old schema.

Version 1:

Reviewer comments:

Reviewer #1

(Remarks to the Author)

I would like to thank the authors for the extensive work conducted to revise the manuscript. The addition of the chemogenetic experiment is nice, even though it does not fully resolve the causality issue. But these data together with the fair discussion

on this make it a reasonable and interesting take overall. All my comments have thus been addressed.

Reviewer #2

(Remarks to the Author)

Thank you for your thorough work during the rebuttal process, especially for adding the new animal data. I have no further issues.

Reviewer #3

(Remarks to the Author)

The authors have adequately addressed all of my comments.

Reviewer #4

(Remarks to the Author)

The authors have satisfactorily addressed my previous concerns regarding causality and have made substantial revisions that improve the overall quality of the paper. I have no additional comments.

REVIEWER COMMENTS

Reviewer #1 (Remarks to the Author):

This work investigates how the orbitofrontal cortex (OFC) contributes to learning new behavioral rules that conflict with prior knowledge. Using a well-designed odor-guided decision task in rats, combined with single-unit electrophysiology and population-level decoding methods, the authors report that OFC neural activity retains representations of previously learned schemas even after a new rule is adopted. Interestingly, the strength of this residual encoding is positively correlated with performance on the new rule, suggesting a facilitative rather than interfering role.

This is a timely and conceptually important study that addresses the neural basis of schema-dependent learning and cognitive flexibility. However, some causal inferences may not be fully supported by the data, and some methodological clarifications are needed to support key claims as outlined below.

We thank the reviewer for recognizing the importance of this study. We have tempered our interpretation to emphasize the correlational nature of the original findings (Fig. 7) and added a new chemogenetic inactivation experiment demonstrating that OFC activity is required for schema generalization and subsequent acquisition of a contradictory rule (new Fig. 8). We also clarified key methodological details, including statistical procedures, classifier validation, and trial selection, to address the points raised.

Major Issues

First, the central claim that residual representations of the prior schema facilitate learning of a new rule is stated in causal terms but rests entirely on correlational evidence. Although a positive relationship between decoder fidelity for the prior rule and behavioral performance on the new rule is reported, there is no experimental manipulation to demonstrate a mechanistic role. It remains possible that both effects arise from a shared third factor, such as general task engagement or learning ability. Without causal disruption (e.g., OFC inactivation or schema-specific interference – note that I am not necessarily calling for more experiments), the authors should temper the interpretation of their results and clarify that the facilitative effect is an inference, not a demonstrated mechanism.

The control group (rats trained only on the cue-identity rule) is certainly helpful in showing that residual non-match encoding depends on prior experience but it still does not directly test whether this residual representation is necessary for accelerated learning. The authors should more clearly acknowledge this limitation when interpreting the putative facilitative role of prior schema encoding. This issue is also more prominent

because of the relatively small sample size (eight male rats) and the lack of detailed reporting on how many neurons were recorded per rat, per session, and per condition. Given the variability in both behavior and neural encoding, it is essential to show that the reported effects are robust across subjects. Cross-animal reliability measures or bootstrapping across animals would help to substantiate the population-level claims.

We thank the reviewer for this important and constructive comment. In the revised manuscript, we now explicitly emphasize that the observed positive relationship between decoder fidelity for the prior rule and behavioral performance on the new rule is correlational in nature. We now clearly acknowledge that this association could reflect the influence of a shared third factor (e.g., task engagement or the activity in different brain area), and that the facilitative effect should therefore be interpreted as an inference rather than a demonstrated mechanism.

To address the reviewer's concern about causality, we performed a new experiment using a chemogenetic approach to transiently inactivate OFC during schema generalization. These data, now presented in Figure 8, show that OFC inactivation produces a prolonged impairment in the ability to subsequently acquire a contradictory schema. While these results do not directly target the specific residual firing patterns highlighted in our original analyses (a limitation we note in the Discussion), they do provide causal support for a mechanistic role of OFC activity in facilitating new learning based on prior schemas.

To strengthen transparency and robustness, we have made several additional revisions. We added a supplemental table reporting the number of neurons recorded per rat, per session, and per condition. The experimental group was also expanded by including one additional animal, thereby increasing the dataset's reliability. In response to the reviewer's request for cross-animal validation, we now present in Supplementary Figure 4 the neural encoding performance plotted separately for each rat, which demonstrates that the reported effects are consistent across individuals. Finally, we conducted a leave-one-subject-out (LOSO) sensitivity analysis of the UMAP projections: for each iteration, population-level embeddings were recomputed while systematically excluding one subject. The resulting low-dimensional structures remained stable across iterations, indicating that the UMAP findings are not driven by any single animal. This analysis is described in the Methods and the results are included in Figure 5d–e, where we report the mean and SEM across LOSO iterations.

Was there a reason to not include female rats in the study?

We thank the reviewer for raising this important point. In our initial pilot experiments, we included female rats and observed no differences in their ability to acquire the task. However, we encountered repeated technical difficulties with electrode implants in females, as the headcaps tended to detach more frequently, we believe due to their

smaller skull size. For this reason, we proceeded with male rats in the main electrophysiological experiments. Importantly, in our new behavioral inactivation experiment, we included female rats and found no differences compared to males in task acquisition or performance. These findings are consistent with previous work which did not find reliable sex differences in OFC-dependent cognitive functions (Hart EE, Sharpe MJ, Gardner MP & Schoenbaum G, *Elife* 9, 2020). We have attached this comparison to the rebuttal and now note it explicitly in the revised manuscript.

Accuracy across different problems under the non-match (blue) and cue-identity (yellow) rules (data are shown as mean \pm SEM with individual data points overlaid). Solid lines indicate males, and dashed lines indicate females. No main effect of sex was observed ($F(1,240) = 3.78$; $p = 0.053$). While the three-way interaction ($\text{Sex} \times \text{Session} \times \text{rule}$) reached nominal significance ($p = 0.012$), post-hoc Tukey tests within each (session \times rule) combination revealed no significant differences between sexes, and the interaction did not remain significant after correction for multiple comparisons (Bonferroni $p > 0.05$). Thus, no consistent main or interaction effects of sex were observed.

Second, for the neural decoding methods, greater transparency and methodological detail may be warranted. It is not clearly stated whether classifiers were validated within sessions, across sessions, or across subjects. Similarly, the use of pseudo-populations across animals raises concerns about overfitting and the independence of the training and test data. The authors should clarify how they ensured that the decoding results reflect meaningful neural structure and not artifacts of trial averaging or classifier bias.

We appreciate this important point and have clarified our methods accordingly. All neural decoding analyses presented in Figures 6, 7, and Supplementary Figure 6 were conducted separately within each animal and each training session; no pseudo-populations were used for these analyses. Pseudo-populations were constructed only for the UMAP visualizations, which were intended to illustrate representational structure across animals rather than to assess decoder performance.

To ensure that decoding results reflect meaningful neural structure rather than artifacts of trial averaging or classifier bias, classifier performance was validated using leave-one-out cross-validation, implemented independently for every rat and every session. In addition, observed classifier accuracies were compared against null distributions generated by shuffling trial labels (1000 permutations per comparison). These details have now been added to the Methods section for clarity.

Additionally, the potential confound between reward contingency and schema representation deserves further consideration, as most decoding contrasts involve rewarded vs. non-rewarded trials that differ along both axes.

We thank the reviewer for this thoughtful comment. To minimize the potential confound between reward contingency and schema representation, all decoding analyses were restricted to the odor-sampling period prior to reward delivery or consumption, thereby isolating neural activity related to rule representation rather than reward receipt. Additionally, we would note that our main finding runs counter to the concern that decoding merely reflects reward outcome, since we find that the stronger the representation of the old rule (i.e., rewarded vs. non-rewarded according to that rule), the more likely the animal was to ignore this distinction and instead act according to the new rule. Thus, rather than reflecting a trivial reward contingency code, the persistence of the old schema representation in OFC predicted its selective gating when incongruent with current task demands. We have clarified this point in the Discussion.

A final concern relates to the interpretation of "multiplexing" in OFC. The idea that OFC simultaneously encodes multiple schema representations is interesting but underspecified. The authors do not provide a clear analysis of whether the same neurons encode both schemas, or whether distinct populations are involved. This distinction is important, both mechanistically and conceptually. Dimensionality reduction or clustering analyses could help clarify whether the old and new rule representations occupy separable neural subspaces or overlap within a shared coding space.

We thank the reviewer for raising this important point. As shown in Figures 4b and 4d, our single-neuron analyses revealed that some neurons responded to both rules, consistent with multiplexed coding at the single unit level. Example PSTHs of such neurons are provided in Supplementary Fig. 3. These findings suggest that at least a subset of OFC neurons carry information about both the prior and the new rule, rather than the representations being strictly segregated into distinct populations.

In addition, we directly addressed the reviewer's suggestion by analyzing the geometry of the population code (new analyses in Fig. 5F–I). Specifically, we examined whether the old and new rule representations occupied overlapping or separable neural subspaces. These analyses revealed that while residual coding of the prior rule

persisted after the switch, the readout axes for the two rules progressively drifted toward near-orthogonality as learning advanced, indicating that OFC represents the two schemas in increasingly distinct subspaces. Thus, although limited single-unit multiplexing is evident, at the population level multiplexing appears to be implemented through parallel, selectively readable codes embedded within largely orthogonal subspaces. We have revised the relevant Results section to explicitly describe this distinction between single-unit and population-geometry organization.

Minor Issues

Several statistical details are not fully described. For example, the authors report many ANOVAs and permutation tests across figures and sessions, but it is not always clear whether corrections for multiple comparisons were applied. Exact p-values, effect sizes, and confidence intervals should be reported consistently, especially for the key correlations between neural decoder accuracy and behavior.

The manuscript is generally well written, though there are instances where the phrasing overstates the results (e.g., “culminating in psychopathology” in the abstract) or introduces ambiguity. Some figure panels are densely labeled, and the font sizes in certain plots are small. Improvements to figure readability (particularly in the decoding and UMAP plots) would enhance accessibility.

The concluding discussion section includes a thoughtful comparison between biological and artificial learning systems, which is intellectually interesting but somewhat tangential. This comparison could be trimmed or better integrated into the main themes of the manuscript.

We thank the reviewer for these constructive suggestions.

Statistical reporting: We have now clarified in the Methods and Results sections that corrections for multiple comparisons were systematically applied to ANOVAs and permutation tests where appropriate. Exact p-values, effect sizes, and 95% confidence intervals are now reported consistently throughout the manuscript and in supplementary tables.

Wording: We revised phrasing that may have overstated the results, including softening the statement in the abstract to avoid implying direct causation with psychopathology. We carefully reviewed the text to ensure accurate and balanced presentation throughout.

Figure readability: We increased font sizes and simplified labeling across several figures to improve clarity, with particular attention to the UMAP plots. To enhance readability, we moved a subset of session UMAP plots to a new supplementary figure that now includes all sessions. Figure captions were also revised for better readability and precision.

Concluding discussion: We streamlined the concluding comparison between biological and artificial learning systems, integrating it more directly with the main themes of schema retention and cognitive flexibility. This keeps the AI discussion aligned with the manuscript's central findings while retaining the conceptual link for broader readership.

Reviewer #2 (Remarks to the Author):

Maor and colleagues use an odor-discrimination task to explore how the orbitofrontal cortex (OFC) represents two competing behavioral rules. Rats were first trained to follow a "non-match" rule (respond when the current odor is different from the previous one) and later learned a "cue identity" rule (respond only to four specific odors, regardless of sequence). Recordings from the lateral OFC showed that both behavior and neural activity adapted with repeated exposures to the task, suggesting schema formation, and after switching rules, most neurons adjusted to the new schema, though a small group continued to encode the old rule. Interestingly, the stronger this persistent activity was, the faster the rats learned the new rule.

The study uses a creative task design and a broad range of analytical tools to explore the data. This work significantly advances our understanding of how the brain supports rule-based decision-making and introduces compelling new insights.

However, I have several concerns, primarily regarding methodological clarity and analytical rigor. These are listed in order of importance

We thank the reviewer for the thoughtful and constructive evaluation of our manuscript.

1. The correlation results in Figure 7 are based on only four data points. While I appreciate the technical and logistical challenges of in vivo electrophysiology, this limited sample raises concerns about the robustness of the findings. I suggest either removing this result or including additional animals to strengthen statistical validity. If this is not possible, these limitations should be explicitly discussed and acknowledged clearly in both in the manuscript and abstract.

We thank the reviewer for this thoughtful comment. In the revised manuscript, we have included data from an additional animal in the experimental group, thereby increasing the sample size for the correlation analysis. We also now emphasize more explicitly in both the Results and Discussion that these findings are correlative in nature and should be interpreted with caution. Importantly, however, we have complemented this analysis with a new chemogenetic inactivation experiment, which provides causal evidence supporting the proposed role of OFC in schema-dependent learning. We hope these

additions address the reviewer's concern and strengthen confidence in the robustness and interpretation of these data.

2. It is unclear whether the analyses included only correct trials. If so, the authors should explicitly detail how trial variability was addressed, particularly concerning trial stratification for analyses like SVM decoding. Differences in trial counts at different training stages could significantly bias decoding results. Conversely, if incorrect trials were included, the authors must clarify how differences in trial correctness between experimental and control groups were accounted for, especially given early discrepancies in 'correct' trial proportions for non-match rule between these groups during initial cue training.

We thank the reviewer for raising this critical clarification. All trials (both correct and incorrect) were included in the decoding analyses, rather than restricting to correct trials only. This approach allowed us to capture the full range of neural activity patterns associated with each rule, including error-related activity, which we considered to be an integral part of learning.

To prevent potential biases due to unequal trial numbers across conditions or training stages, classifiers were validated using leave-one-out cross-validation separately for each rat and each session. This procedure ensures that each trial is tested against a model trained on all other trials within the same session, thereby minimizing the risk of overfitting and mitigating imbalances across sessions.

We acknowledge that there were differences in the proportion of correct versus incorrect trials across training stages, as expected during learning. To address this, we confirmed that decoding performance remained significantly above chance when compared to shuffled-label control distributions (1000 permutations per comparison). Thus, the results cannot be explained by trial count differences or biases in classifier performance. We now clarify these methodological points in the Methods and Results sections.

3. For results shown in Figure 5e, the statistical comparison should be directly conducted between control and experimental groups rather than indirectly through comparisons to shuffle conditions. The authors could, for instance, test whether the difference between experimental and control conditions is significantly greater than differences observed between two null distributions.

We thank the reviewer for this constructive suggestion. Following it, we have now directly compared the difference between the experimental and control groups against the distribution of differences obtained from shuffled data. This analysis has been added

to the Methods and to the relevant Results section, and the outcomes are now reported in the revised manuscript.

At the same time, we believe that the comparison of each group to its respective shuffle remains informative, as it provides a direct estimate of how likely the observed distances are to occur by chance. This, in turn, offers an important measure of the residual representational structure within each group, independent of the group comparison. We have therefore retained both sets of analyses in the manuscript.

4. The analysis presented in Figure 6g lacks clarity regarding the experimental phase from which data were derived. Again, given that early cue identity training produces notable differences in correct trial proportions between groups (experimental and control), the presented analysis could potentially be biased by these initial performance disparities.

We thank the reviewer for raising this important concern. Figure 6g shows the ratio of the main diagonal to the side diagonal of the confusion matrix for each rat across all sessions of cue-identity learning (A and B problems). We agree that early cue-identity training could in principle bias this analysis due to initial performance disparities. However, we do not find evidence for such disparities with respect to the relevant cue-identity rule. As shown in Figure 2g (yellow curves, correct ratios for the cue-identity rule), there are no significant differences between groups at any stage of cue-identity learning.

What we do observe is that, in the early stages, neural representations remain biased toward the previously learned non-match rule. This is reflected also in a higher main-to-side diagonal ratio during these sessions. To rule out the possibility that differences in correct performance according to the non-match rule could account for this effect, we repeated the analysis after excluding the first two stages of cue-identity training from both groups. Even after this exclusion, we still observed a significant group difference both in an ANOVA across stages (Figure A) and in a direct comparison as in Figure 6g

(Figure B). We now clarify this point in the manuscript.

A. Impact of the non-match rule on population representation during cue-identity sessions, excluding the first two stages, calculated as the ratio between probabilities in the main diagonal and side diagonals of the confusion matrices, for each session. Ratio in the experimental group was significantly higher than in the control ($F(1,53) = 4.62, p = 0.036$). B. Main/side diagonal ratio over all sessions excluding the first two stages. (Mann–Whitney U test, $U = 766, p = 0.029$).

5. The authors employ a mix of parametric, non-parametric, and permutation statistical methods. It would be beneficial to explicitly justify the rationale behind selecting these various statistical approaches throughout the manuscript.

We appreciate the reviewer's attention to this point. In the revised manuscript, we have added explicit justifications for our choice of statistical approaches throughout the Methods sections. Our selections were guided by standards in the field and by the characteristics of the data in each analysis. Importantly, the main findings were robust across multiple approaches and remained significant regardless of the specific test applied. If there are particular results the reviewer would like to see reanalyzed with an alternative method, we would be happy to provide these additional analyses.

6. The text results associated with Figure 6f is unclear, as this panel contains two separate plots. The authors should indicate whether the reported statistical results pertain to both plots or only one.

The reported statistical results in Figure 6f apply to both problems in the cue-id phase. We have clarified this explicitly in the text.

7. Figure 3C Adding session numbers to the x-axis of Figure 3c would improve plot readability and facilitate interpretation.

We have added session numbers to the axis of this Figure.

Reviewer #3 (Remarks to the Author):

This manuscript presents exciting and novel findings demonstrating that orbitofrontal cortex (OFC) neural representations track rule learning and, as hypothesized, both OFC activity and task performance were facilitated by prior learning of a different rule. A particularly important and unexpected result was that OFC ensembles continued to represent the previously learned rule even after rats acquired the new one—and more strikingly, the stronger the OFC representation of the old cue, the better the performance after the rule switch. These findings provide compelling evidence that persistent encoding of prior schemas in the OFC may actively support, rather than hinder, behavioral flexibility. The results will be of broad interest to neuroscientists studying cognition, learning, and prefrontal function. While the manuscript is strong overall, there were some minor issues with clarity, including a somewhat niche chess analogy, a lack of explanation for the fixed rule order, and the need for clearer figure presentation and labeling—particularly in Figures 2I and 3B/C. Moreover, one limitation was that only male rats without justification.

The work will be of significance to the field because it advances our knowledge of how OFC neural ensembles represent new and old information, adding empirical evidence to the research on schemas. Some additional literature review that mentions the work on schemas and memory consolidation in the hippocampus could be relevant here even though the manuscript focuses on OFC and rule switching.

See <https://doi.org/10.1126/science.1135935>

We thank the reviewer for their supportive comments and for highlighting the relevance of schema-related literature. Indeed, our work was strongly inspired by studies on schema and memory consolidation in the hippocampus. The review suggested (Tse et al., 2007, Science; doi:10.1126/science.1135935) was already cited in the Introduction. To further strengthen the connection, we have now added additional references (Bakermans et al., 2025; Ma et al., 2025; Sommer et al., 2022; Stachenfeld et al., 2017; Xiao et al., 2025) that discuss the role of the hippocampus in schema learning and memory consolidation, as well as its interaction with OFC.

The conclusion, that the OFC contributes to new learning by representing the old rule as the new rule is learned, is strongly supported by the data.

Only minor issues are noted with the clarity of the presentation. These are:

-The chess analogy could be made more general. Some of the specifics will only be understandable to readers who are familiar with chess strategy.

In line with the comment, we revised the Introduction to replace the chess analogy with a more broadly accessible driving example. We describe how entering a country with different traffic conventions (e.g., side of the road, right-of-way at roundabouts, sign meanings) initially triggers home-country schema, producing systematic slips until the new rule set is learned and the prior schema is suppressed. This example illustrates both the value and the limits of schemas in everyday behavior and should be readily accessible to a wide readership.

We have moved the more domain-specific chess and Go analogy to the Discussion, where it is now presented in the context of artificial intelligence. Because learning these two games is a canonical example in AI research, this placement provides a natural and relevant connection between human schema use, OFC function, and computational approaches to learning.

-The authors should explain why they did not counterbalance the order of the rules. All rats learned the nonmatch rule first, followed by the cue-id rule. Was there some practical reason for this? What would happen if rats learned the cue-id rule first, then the nonmatch rule. Some of this concern is mitigated by the control group that learned only the cue-id rule. I am not suggesting that the study design was flawed. Instead, I would like the authors to explain why they didn't employ this approach which is very common in the field of behavioral neuroscience.

We appreciate the reviewer's insightful comment. We did not counterbalance rule order because, in pilot work, rats that learned the cue-id rule first were markedly slower and more variable when subsequently acquiring the non-match rule. This led to substantially prolonged training. Importantly, this would have made a counterbalanced design both impractical and difficult to compare to the current dataset due to large differences in total training exposure and trial counts across groups. To avoid these confounds and preserve statistical power and comparability, we adopted a fixed order in which all animals learned the more demanding non-match rule before cue-id. We now explicitly acknowledge the fixed order as a design limitation in the Discussion.

-For Fig. 2, Panel I, the bars and horizontal lines did not come through on the pdf, making the figure legend and caption confusing.

We have fixed this issue with the display of this figure.

-Fig. 3B: More details are needed. How were the units ordered in the z-score grayscale map?

For each single unit, z-scores were calculated separately for rewarded and non-rewarded trials by normalizing the mean firing rate in the response window (500 ms prior to odor un-poke) against baseline activity (1 s prior to trial initiation). The z-score for non-rewarded trials was then subtracted from that of rewarded trials to obtain the delta z-score, representing the differential standardized activity between trial outcomes. Units were subsequently ordered by their latency to peak activity, and the colormap was trimmed between -2 and 2 to enhance visibility. These details have now been added to the Methods section and clarified in the Figure 3B caption.

-Fig. 3C: Please clarify what is being shown in the rightmost plot. The title and legend are the same as the leftmost plot and the caption does not give any clues.

We thank the reviewer for catching this. The rightmost plot in Figure 3C represents the second learning sessions of the nonmatch rule A problem (A'), in which rats were retested on the original problem. The title mistakenly read (A) instead of (A'). This has now been corrected, and we clarified the distinction explicitly in the figure caption.

The methodology is sound, except there is no justification for using only male rats. Single-sex (male-only) studies are becoming increasingly rare in science, especially with no justification included. In fact, most top tier journals as well as federal funding agencies are requiring not only inclusion of both sexes, but that sex be analyzed as a variable.

In our initial pilot experiments, we included female rats and observed no differences in their ability to acquire the task. However, we encountered repeated technical difficulties with electrode implants in females, as the headcaps tended to detach more frequently, we believe due to their smaller skull size. For this reason, we proceeded with male rats in the main electrophysiological experiments. Importantly, in our new behavioral inactivation experiment, we included female rats and found no differences compared to males in task acquisition or performance. These findings are consistent with previous work showing no reliable sex differences in OFC-dependent cognitive functions (Hart EE, Sharpe MJ, Gardner MP & Schoenbaum G, *Elife* 9, 2020). We have attached this comparison to the rebuttal and now note it explicitly in the revised manuscript.

There is enough detail included in the methods for the work to be reproduced.

Reviewer #4 (Remarks to the Author):

This manuscript by Maor et al. investigates the role of the orbitofrontal cortex (OFC) in supporting flexible learning when confronted with conflicting schemas. Through a combination of behavioral experiments, single-unit neural recordings, and advanced population decoding analyses, the study reveals that the OFC retains representations of prior schemas while concurrently encoding new ones. This dual representation provides a framework for flexible adaptation to environmental demands and highlights how the brain resolves conflicts between old and new behavioral rules.

The core finding is that prior schemas are not overwritten but persist in the OFC even during the acquisition of conflicting schemas. This challenges traditional theories of interference and forgetting and offers an alternative neural mechanism underlying cognitive flexibility.

I like the paper a lot in general, but with one major concern. While the authors convincingly demonstrate that animals with stronger representations of prior schemas perform better on new tasks, it is not entirely clear whether this relationship is causal. An alternative explanation could be that these animals are simply better learners overall, independent of the persistence of the old schema. Addressing this potential confounder would strengthen the authors' interpretation. For instance, maybe the authors could look at the baseline learning ability or conduct experiments to selectively manipulate the representation of the old schema.

We appreciate the reviewer's thoughtful comment and agree that our initial findings were correlative in nature. In the revised manuscript, we now explicitly emphasize that the positive relationship between decoder fidelity of the prior rule and behavioral performance on the new rule is correlational in nature. Specifically, we now note in the discussion that the relationship is correlational and could reflect that both the behavior and the firing pattern in OFC are driven by some common "third factor". Following the reviewer's suggestion, we also performed an additional analysis to test whether the observed relationship could simply reflect general differences in learning ability across animals. Notably, no significant correlation was found between performance on the initial non-match rule and performance on the subsequent cue-identity rule (Fig. 7f; Pearson correlation, all $p > 0.1$). Thus, animals that learned the first rule more efficiently did not necessarily acquire the second rule faster, indicating that the relationship cannot be explained by a superior baseline learning ability.

Additionally, to address the concern about the causal role of OFC in the behavior, we conducted a new experiment using a chemogenetic approach to transiently inactivate OFC during schema generalization. The results, now presented in Figure 8, demonstrate that OFC inactivation has a prolonged impact on the ability to subsequently acquire a contradictory schema. While it is difficult to specifically address the causal role of the specific pattern of firing we identify (a limitation we note in the Discussion), we believe these findings do provide causal support for a mechanistic role of OFC in facilitating new learning based on prior schemas.

Bakermans, J. J. W., Warren, J., Whittington, J. C. R., & Behrens, T. E. J. (2025). Constructing future behavior in the hippocampal formation through composition and replay. *Nat Neurosci*, 28(5), 1061-1072. <https://doi.org/10.1038/s41593-025-01908-3>

Ma, F., Lin, H., & Zhou, J. (2025). Prediction, inference, and generalization in orbitofrontal cortex. *Curr Biol*, 35(7), R266-R272. <https://doi.org/10.1016/j.cub.2025.02.021>

Sommer, T., Hennies, N., Lewis, P. A., & Alink, A. (2022). The Assimilation of Novel Information into Schemata and Its Efficient Consolidation. *J Neurosci*, 42(30), 5916-5929. <https://doi.org/10.1523/JNEUROSCI.2373-21.2022>

Stachenfeld, K. L., Botvinick, M. M., & Gershman, S. J. (2017). The hippocampus as a predictive map. *Nat Neurosci*, 20(11), 1643-1653. <https://doi.org/10.1038/nn.4650>

Xiao, Z., Wang, X., Zhang, J., Ou, J., He, L., Qu, Y., Hu, X., Behrens, T. E. J., & Liu, Y. (2025). Human hippocampal ripples align new experiences with a grid-like schema. *Neuron*. <https://doi.org/10.1016/j.neuron.2025.07.028>